# Primary cilia deficiency in neural crest cells models anterior segment dysgenesis in mouse

Céline Portal[1], Panteleimos Rompolas[2], Peter Lwigale[3], Carlo Iomini[1,4]*

[1]Department of Ophthalmology, Wilmer Eye Institute, Johns Hopkins University School of Medicine, Baltimore, United States; [2]Department of Dermatology, Institute for Regenerative Medicine, Perelman School of Medicine, University of Pennsylvania, Philadelphia, United States; [3]BioSciences Department, Rice University, Houston, United States; [4]Department of Cell Biology, Johns Hopkins University School of Medicine, Baltimore, United States

**Abstract** Defects affecting tissues of the anterior segment (AS) of the eye lead to a group of highly debilitating disorders called Anterior Segment Dysgenesis (ASD). Despite the identification of some causative genes, the pathogenesis of ASD remains unclear. Interestingly, several ciliopathies display conditions of the AS. Using conditional targeting of *Ift88* with *Wnt1-Cre*, we show that primary cilia of neural crest cells (NCC), precursors of most AS structures, are indispensable for normal AS development and their ablation leads to ASD conditions including abnormal corneal dimensions, defective iridocorneal angle, reduced anterior chamber volume and corneal neovascularization. Mechanistically, NCC cilia ablation abolishes hedgehog (Hh) signaling in the periocular mesenchyme (POM) canonically activated by choroid-secreted Indian Hh, reduces proliferation of POM cells surrounding the retinal pigment epithelium and decreases the expression of *Foxc1* and *Pitx2*, two transcription factors identified as major ASD causative genes. Thus, we uncovered a signaling axis linking cilia and ASD.

*For correspondence:
ciomini1@jhmi.edu

Competing interests: The authors declare that no competing interests exist.

## Introduction

Anterior segment dysgenesis (ASD) is a term referring to a spectrum of congenital disorders of the anterior segment (AS) structures of the eye. Abnormalities typically associated with ASD include; corneal opacity, cataract, posterior embryotoxon, iris hypoplasia, corectopia or polycoria, and adhesions between the iris and cornea or lens and cornea (*Reis and Semina, 2011*). Approximately 50% of ASD patients develop glaucoma that can lead to visual impairment and blindness (*Cvekl and Tamm, 2004*; *Ito and Walter, 2014*; *Ma et al., 2019*; *Reis and Semina, 2011*). Although mutations in genes encoding transcription factors, transporters, and glycosylating proteins have been described in patients with ASD, several ASD cases still await genetic elucidation (*Cvekl and Tamm, 2004*; *Ito and Walter, 2014*; *Ma et al., 2019*; *Reis and Semina, 2011*), and the cellular and molecular mechanisms underlying the pathogenesis of different ASD conditions remain unknown. During development, most AS structures are derived from the neural crest cells (NCC), including the corneal stroma and endothelium, ciliary body muscle and body, iris stroma, and the trabecular meshwork (*Gage et al., 2005*). Migrating NCC of the periocular mesenchyme (POM) begin to invade the AS of the eye between E11.5 and E12.5 during murine development and differentiate into corneal endothelial cells and keratocytes (*Pei and Rhodin, 1970*; *Swamynathan, 2013*). By E16.5, the presumptive iris is visible and detaches from the cornea, while the drainage structures (trabecular meshwork and Schlemm's canal) continue to develop postnatally (*Gould et al., 2004*). NCC also gives rise to the sclera, pericytes of the choroid and hyaloid vasculature, orbital cartilage and bone, and

oculomotor tendons (*Cavodeassi et al., 2019*). The neural crest is a transient embryonic structure in vertebrates that delaminates from the border between the neural plate and the non-neural ectoderm; NCC migrate throughout the embryo to multiple locations and differentiate into a wide variety of cell types and tissues (*Mayor and Theveneau, 2013*; *Sauka-Spengler and Bronner-Fraser, 2008*). Recent studies have shown that primary cilia play essential roles in morphogenetic processes involving neural crest-derived cells such as craniofacial development (*Chang et al., 2015*; *Schock and Brugmann, 2017*). It has been proposed that primary cilia mediate tissue-tissue interactions requiring reciprocal signaling rather than purely NCC specification (*Brugmann et al., 2010*; *Tobin et al., 2008*).

Primary cilia are microtubule-based cellular organelles that emanate from the basal body and extend from the plasma membrane. A bidirectional movement of protein particles along the axoneme, called intraflagellar transport (IFT), ensures the appropriate assembly and maintenance of cilia (*Iomini et al., 2001*; *Ishikawa and Marshall, 2011*; *Kozminski et al., 1995*; *Rosenbaum and Witman, 2002*; *Taschner and Lorentzen, 2016*). Primary cilia play a pivotal role in the development and tissue homeostasis by regulating multiple signaling pathways, most notably Hedgehog (Hh) (*Bangs and Anderson, 2017*; *Reiter and Leroux, 2017*), and also Wnt, Notch and others (*May-Simera et al., 2017*; *Pala et al., 2017*). Upon cilia binding to one of the three mammalian Hh ligands (Sonic hedgehog (SHH), Indian hedgehog (IHH) or Desert hedgehog (DHH)) to the receptor Patched (PTCH), the Hh transducer Smoothened (SMO) accumulates in the cilia and PTCH is excluded. Subsequently, SMO activates the Gli transcription factors which translocate to the nucleus and activate the expression of Hh target genes (*Elliott and Brugmann, 2019*) involved in cell proliferation, maintenance of stemness, cell-fate determination, cell survival and epithelial to mesenchymal transition (*Katoh and Katoh, 2009*).

A wide range of human diseases can be attributed to dysfunctions of cilia, known as ciliopathies, which affect most human organ systems, including the AS of the eye (*Reiter and Leroux, 2017*). Patients affected by Meckel syndrome, a severe ciliopathy, present AS abnormalities including microphthalmos/anophthalmos, aniridia, cryptophthalmos, sclerocornea, abnormal corneal thickness, corneal neovascularization, and abnormal iridocorneal angle (*MacRae et al., 1972*). Oral-facial-digital syndrome has recently been reported to include several cases of microphthalmia (*Abuhamda and Elsous, 2018*; *Cortés et al., 2016*) and a patient affected by Joubert syndrome has been described with corneal opacity (*Zaki et al., 2011*). More commonly, glaucoma and cataract are conditions associated with Bardet-Biedl and Lowe syndromes (*Eibschitz-Tsimhoni, 2003*; *Luo et al., 2014*), cataract and keratoconus with Leber's congenital amaurosis (*Eibschitz-Tsimhoni, 2003*), and one of the features of the Biedmond syndrome type 2, resembling to the Bardet-Biedl syndrome, is the presence of iris colloboma (OMIM #210350) (*Schachat and Maumenee, 1982*).

Several studies focusing on craniofacial development have demonstrated that ablation of the primary cilium via inactivation of the IFT in NCC leads to severe craniofacial defects (*Brugmann et al., 2010*; *Chang et al., 2016*; *Millington et al., 2017*; *Snedeker et al., 2017*; *Tian et al., 2017*; *Watanabe et al., 2019*). Interestingly, systemic mutations in ciliogenic genes causing an increase in Hh activity and a deletion of *Gli3*, which acts predominantly as a repressor of the Hh target genes, lead to similar abnormal ocular development (*Burnett et al., 2017*; *Franz and Besecke, 1991*; *Furimsky and Wallace, 2006*; *Johnson, 1967*; *Wiegering et al., 2019*). Conversely, loss or downregulation of Hh activity leads to severe craniofacial and ocular defects including anophthalmia, cyclopia, microphthalmia, and coloboma (*Cavodeassi et al., 2019*), leading us to hypothesize that the primary cilium plays a pivotal role in the development of the AS and that dysfunction of the primary cilium in NCC could lead to conditions similar to those associated with ASD.

In this study, we aimed to investigate the role of the primary cilium in the development of NCC derived ocular structures and its possible role in ASD. We showed that NCC of the POM are ciliated and that primary cilia persist in keratocytes of adult mice. By ablating primary cilia in NCC, we were able to induce an ASD phenotype with impaired corneal stroma organization and dimension, abnormal iridocorneal angle and corneal neovascularization. We also observed a reduction of the Hh signaling pathway and the cell proliferation specifically in a subset of cells in the POM surrounding the retinal pigment epithelial cells (RPE), and a decreased expression of *Foxc1* and *Pitx2*, two transcription factors identified as major ASD causative genes. Furthermore, we identified the endothelial cells of the choroid as the cells expressing *Ihh*, which is the Hh ligand maintaining the Hh activity in the POM surrounding the RPE layer in normal eye development.

## Results

### Primary cilium ablation in NCC leads to ASD phenotype

In order to categorize the role of primary cilia in the development of NCC-derived tissue of the POM we first determined spatiotemporal distribution of primary cilia during murine development by immunofluorescence (IF) using an anti-Arl13b Ab, a widely accepted ciliary marker (*Caspary et al., 2007*) in genetically labeled NCC. NCC were traced by using the *Rosa26^mT/mG* (mT/mG) reporter mouse line (*Muzumdar et al., 2007*) crossed to the *Wnt1-Cre* mouse (*Danielian et al., 1998*) in which Cre recombinase is under the control of the *Wnt1* promoter, a gene highly expressed in early stages of NCC specification. In the *Wnt1-Cre;mT/mG* transgenic line, the Cre-dependent excision of a cassette expressing the red-fluorescent membrane-targeted tdTomato (mT) drove the expression of a membrane-targeted green fluorescent protein (mG) in bona fide NCC-derived tissues (*Figure 1A*). We observed that at E14.5, all NCC-derived tissues of the POM and the presumptive corneal stroma were ciliated (*Figure 1A*).

Our previous studies reported that while primary cilia are present in developing corneal endothelium (also a NCC-derived tissue), they disassemble in adult corneal endothelium at steady state (*Blitzer et al., 2011*). To assess the presence/absence of primary cilia in adult corneas we utilized a transgenic mouse line expressing the ciliary membrane protein somatostatin receptor three fused to GFP under the ubiquitous promoter for actin (Sstr3::GFP) (*O'Connor et al., 2013*). Intravital microscopy revealed that cilia were present in all keratocytes of the corneal stroma of 3-month-old mice (*Figure 1B*). Thus, despite a common embryonic origin with the corneal endothelium, keratocytes maintained cilia into adulthood. To gain ultrastructural insights we analyzed corneal stroma and POM in developing eyes. TEM showed that in developing eyes, cilia emanated from the cellular surface into the extracellular matrix, whereas cilia of newborn keratocytes appeared to be intracellular or largely invaginated in a long ciliary pocket with their axis parallel to the cell plane (*Figure 1D–E*). Interestingly, the tip of cilia in developing cornea and POM were observed to interact with cellular protrusions of neighboring cells (*Figure 1C–D*). Moreover, the plasma membrane of these cellular protrusions at the contact point with ciliary tips appeared to be highly electron-dense, suggesting the presence of protein components or modified lipids in this region (*Figure 1D–E*).

In order to determine if primary cilia are involved in the development of AS we set out to ablate *Ift88*, a subunit of the IFT machinery required for cilia assembly and maintenance (*Pazour et al., 2000*) in NCC. We generated the *Wnt1-Cre;Ift88^fx/fx* mouse (cKO) which was phenotypically indistinguishable from the null hemizygous *Wnt1-Cre;Ift88^fx/-*. In this mouse the *Ift88* gene is excised in all migrating mesenchymal cells expressing *Wnt1* leading to complete ablation of the primary cilium (*Figure 1—figure supplement 1*) (*Chai et al., 2000*; *Danielian et al., 1998*). To monitor ablation of cilia in the NCC of the POM we generated the *Wnt1-Cre;Ift88^fx/fx;mT/mG* mouse and labeled cilia with an anti-Arl13b Ab (*Caspary et al., 2007*). In control mice (*Wnt1-Cre;Ift88^fx/+;mT/mG*), virtually all the NCC of the POM appeared ciliated during development (*Figure 1A*) while in *Wnt1-Cre; Ift88^fx/fx;mT/mG* cKO mice, cilia were absent in most of the POM cells expressing Cre (*Figure 1—figure supplement 1B*). We confirmed ablation of cilia in keratocyte precursors by TEM and observed the basal body apparatus to reach the apical plasma membrane in both control and cKO corneas, however, primary cilia were only observed emanating from basal bodies of control keratocytes (*Figure 1—figure supplement 1C*). *Wnt1-Cre;Ift88^fx/fx* mutant mice died at birth and E18.5 embryos displayed strong craniofacial defects including increased frontal width, wider frontonasal prominence and increase of the distance between the nasal pits, consistent with previous studies (*Figure 2A*) (*Tian et al., 2017*; *Watanabe et al., 2019*). In addition, we detected abnormalities in the anatomy of the eye. At E14.5, a developmental stage preceding the closure of the eyelids, the axis of the eyeballs was misaligned facing downward in cKO whereas, in control it remained perpendicular to the sagittal plane of the head (*Figure 2A*). Furthermore, the outline of the presumptive iris appeared irregular in cKO while in the control it described a nearly perfect circle (*Figure 2A*). Later in development (E18.5), the irregularities of the iris were exaggerated in cKO and the developing cornea appeared smaller than that of the control, strongly arguing toward severe defects of the AS including a reduced anterior chamber (*Figure 2*). Thus, these macroscopic anatomical abnormalities suggest a morphogenetic condition of the AS in the eye of the cKO mice.

To further characterize the mutant phenotype of the AS, we conducted histological analysis of paraffin and plastic embedded samples (*Figure 2*). The eye field of cKO embryos at E10.5 and 14.5

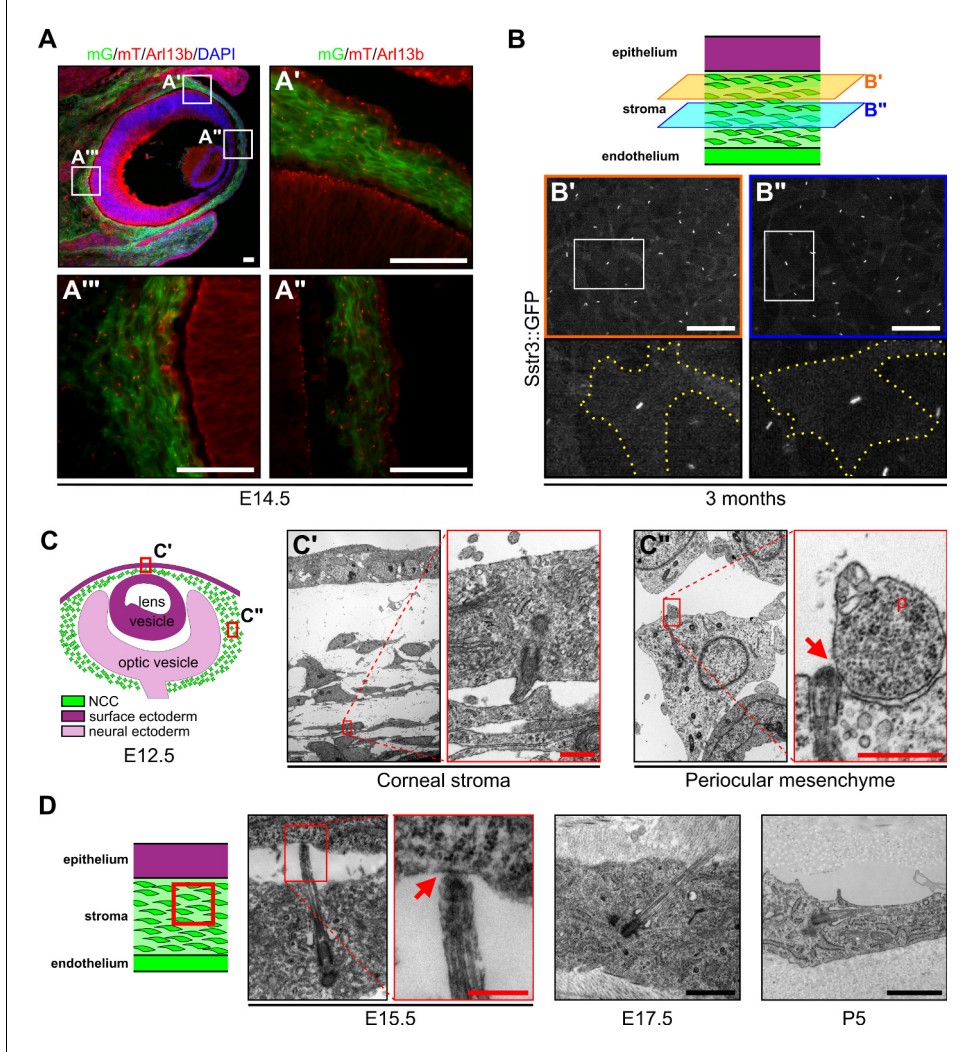

**Figure 1.** NCC of the periocular mesenchyme are ciliated. (**A**) Representative eye section of a *Wnt1-Cre;Ift88*^fx/+^; *mT/mG* embryo at E14.5. NCC express the mG reporter (green cells) whereas cells from other embryonic origin express the mT reporter (red cells). Primary cilia were stained with an anti-Arl13b Ab and appear as small red rods. Scale bar, 50 µm. (**B**) Representative corneal stroma images of an Sstr3::GFP mouse at 3 months, in which primary cilia are GFP fluorescent. All stroma keratocytes are ciliated at adulthood. Scale bar, 50 µm. (**C**) Representative images of primary cilia in the corneal stroma and the periocular mesenchyme at E12.5. Scale bar, 0.5 µm. (**D**) Representative images of primary cilia in the corneal stroma at E15.5, E17.5, and P5. Scale bar, 1 µm. Primary cilia interact with neighboring cells or their cytoplasmic protrusions (red arrows). p, cytoplasmic protrusion.

The online version of this article includes the following figure supplement(s) for figure 1:

**Figure supplement 1.** Genetic deletion of *Ift88* in NCC leads to primary cilium ablation in NCC.

appeared indistinguishable from that of the control (*Figure 2B*). In contrast, at E15.5, we observed a significant reduction of the anterior chamber in the cKO samples (*Figure 2B–D*). At this stage, the mutant was lacking most of the mesenchymal cells condensing at the developing iridocorneal angle between the cornea and the presumptive iris that was clearly visible in the control (*Figure 2D*). As a result, the iridocorneal area appeared disorganized in the mutant with a narrower angle than in the control. Between E16.5 and E18.5, the corneal stroma thickness and the corneal diameter were both significantly reduced in the cKO embryo compared to the control (*Figure 2B–C*). Moreover, the iridocorneal angle abnormalities found in the mutant eye persisted with the presumptive iris significantly shorter than that of the control (*Figure 2D*). Plastic cross sections of the cornea revealed that the density of keratocytes was significantly higher in the cKO stroma compared to control however,

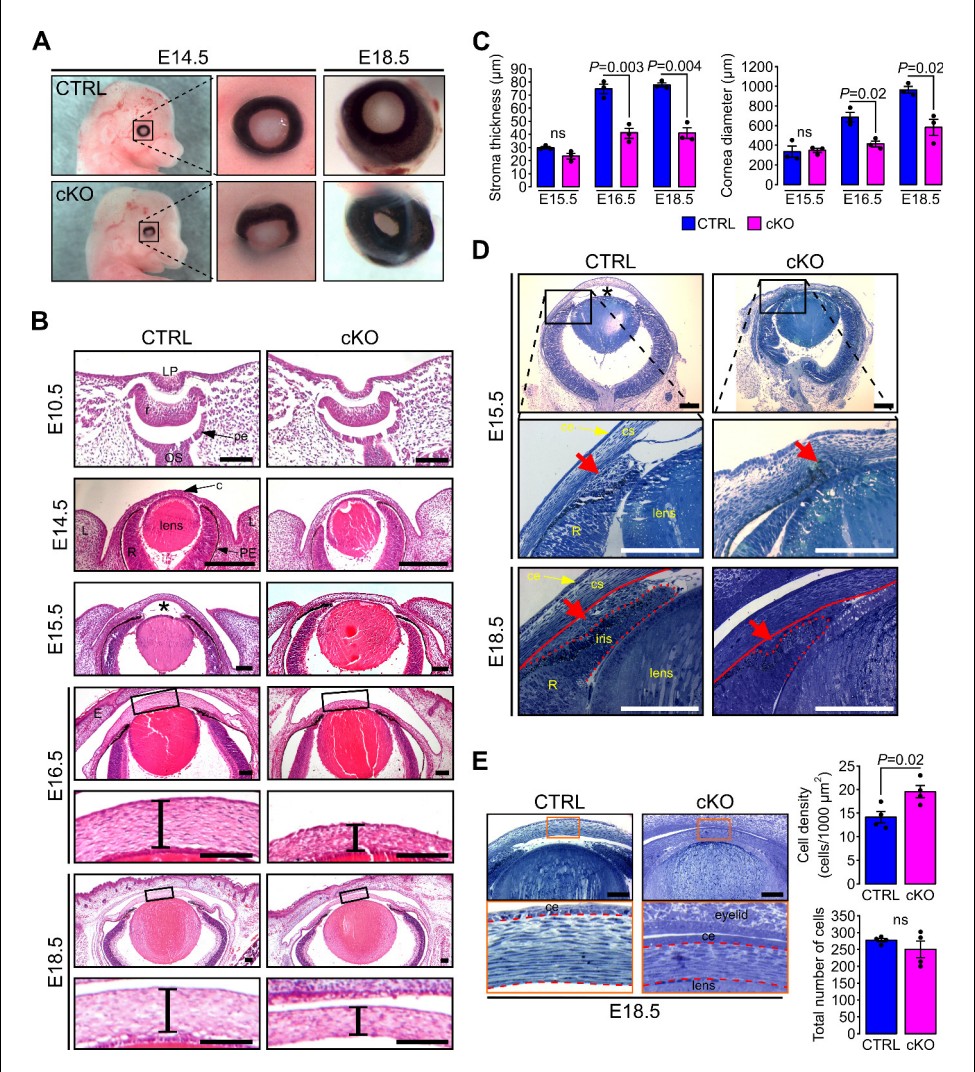

**Figure 2.** Primary cilium ablation in NCC leads to ASD phenotype. (**A**) Representative images of control and cKO eyes at E14.5 and E18.5. Enlarged images show the misorientation of the eyeball in the cKO embryos at E14.5 and the reduced size of the cornea at E18.5. (**B**) Representative eye sections stained with HE of control and cKO embryos at E10.5, E14.5, E15.5, E16.5, and E18.5. Boxed regions indicate the areas shown at higher magnification below. Scale bar, 100 µm; *, anterior chamber; c, cornea; E, fused eyelids; L, lid fold; LP, lens pit; OS, optic stalk; pe, future pigment epithelium; PE, pigment epithelium; r, future neural retina; R, neural retina. (**C**) The corneal stroma thickness and the corneal diameter were measured in the center of the cornea on HE stained paraffin sections at E15.5, E16.5, and E18.5 (n = 3 embryos/group). (**D**) Representative plastic sections of the iridocorneal angle of control and cKO embryos at E15.5 and E18.5. Red arrows show the accumulation of mesenchymal cells between the cornea (underlined by the red line) and the presumptive iris (surrounded by the red dotted line). Scale bar, 150 µm; *, anterior chamber; ce, corneal epithelium; cs, corneal stroma; R, neural retina. (**E**) Representative plastic sections of corneas of control and cKO embryos at E18.5. Boxed regions indicate the areas shown at higher magnification below. Keratocytes were counted in the stroma which is surrounded by red dotted lines. The keratocyte density is significantly increased in cKO embryos compared to control but not the total number of cells (n = 4 embryos/group). Scale bar, 100 µm; ce, corneal epithelium. Data are presented as mean SEM. Statistical significance was assessed using two-tailed Student's *t*-test. ns, non-significant, p≥0.05.

the total number of cells in corneal stroma remained unchanged in both genotypes (*Figure 2E*). The increased keratocyte density was therefore attributed to the smaller volume cornea in cKO embryos compared to control. The unchanged number of keratocytes in the stroma of both genotypes suggests that primary cilium ablation did not impair the NCC migration into the corneal stroma. Thus,

the ablation of the primary cilium in the NCC leads to ASD that is not due to NCC migration defects.

Because of the differences in cornea dimensions and stroma cell density, we examined the 3D organization and the morphology of keratocytes in the mutant and control corneas. Keratocyte morphology and spatial distribution across the stroma vary in mammals, with an increased density, flattening, and extension of the keratocytes in the posterior part of the stroma in late embryonic development (*Haustein, 1983*; *Poole et al., 2003*). To determine stromal organization, we developed a quantitative tool using live confocal imaging on mT/mG mice. Segmentation of the stromal extracellular spaces was carried out with Ilastik (*Haubold et al., 2016*) to then quantify the amount (%) and the average size of extracellular spaces using Fiji (*Schindelin et al., 2012*) (*Figure 3—figure supplement 1*). In control embryos, the amount of extracellular spaces was significantly lower in the posterior than the anterior portion of the corneal stroma, defining an antero-posterior gradient of keratocyte density (*Figure 3B*). This was mirrored in the stroma in control embryos with the size of the extracellular spaces in the posterior part being significantly smaller than in the anterior part, defining an antero-posterior gradient of the extracellular space size (*Figure 3C*). In cKO embryos, the antero-posterior gradient of the keratocyte density was maintained as in control, while the average size of extracellular spaces was not (*Figure 3B*). The amount of extracellular space was significantly reduced in both the anterior and posterior parts of the cKO stroma compared to control (*Figure 3B–C*) making the keratocyte distribution in the corneal stroma of the cKO embryos denser than that of the control. Differences in the anterior-posterior extracellular space gradient and the average size of single extracellular spaces were abnormal in the cKO suggesting a defective tridimensional organization of the keratocytes in the mutant. Thus, the ablation of the primary cilium in NCC impairs the spatial organization of keratocytes in the stroma independently from cell migration.

Because the density and spatial organization of the keratocytes were abnormal in the mutant stroma we sought to investigate the morphology of single keratocytes in vivo. Keratocytes are characterized by their cytoplasmic processes interconnected with each other to form a dense and complex 3D network (*Nishida et al., 1988*). We focused on the junction between the corneal epithelium and the stroma because here the density of keratocytes is lowest. In both genotypes, we distinguished single cytoplasmic processes of the first keratocyte layer expressing the mG reporter underneath the corneal epithelial cell layer expressing the mT reporter. The number of cytoplasmic processes was significantly increased in the cKO embryos compared to control (*Figure 3D–E*), suggesting a possible role of the primary cilium in controlling the morphology and the number of cytoplasmic processes of keratocytes.

## Ablation of Ift88 in NCC disrupts Hh signaling in a subpopulation of POM cells surrounding RPE and at the iridocorneal angle

Morphological changes, as well as spatial organization of the cells in a tissue, occur as cells differentiate. This implies a wide variety of signaling pathways occurring in a timed and coordinated fashion. Mice lacking heparan sulfate in NCC display ASD phenotypes resembling those caused by the lack of NCC cilia including cornea stroma hypoplasia, dysgenesis of the iridocorneal angle and decreased depth of the anterior chamber (*Iwao et al., 2009*). As the TGFβ2 pathway is disrupted in these mice, we assessed whether primary cilium ablation in the NCC affected the TGFβ2 pathway in the cornea. At E18.5, the expression of *Tgfbr1*, *Tgfbr2*, *Smad2*, *Smad3*, *Smad4* and *Smad7* and the percentage of pSmad2/3$^+$ cells in the cornea remained indistinguishable between the cKO and control mice (*Figure 3—figure supplement 2*).

Hh, Wnt and Notch signaling pathways have been linked to the primary cilium (*Pala et al., 2017*). We therefore used quantitative real-time PCR (RT-qPCR) on isolated ocular NCC and reporter mouse lines to examine the expression of specific target genes of the above-mentioned pathways. To isolate ocular NCC, cKO and control mice were crossed to the mT/mG transgenic reporter line (*Figure 1—figure supplement 1*) and mG$^+$ cells of the dissected sclera and cornea tissue were digested and sorted by FACS following the approach illustrated in *Figure 4A*. NCC from E18.5 cKO mouse showed a significant decrease of Hh target genes including *Gli1* and *Ptch1* but not *CyclinD* when compared to control NCC (*Figure 4B*). In contrast, expression of target genes of the Wnt and Notch pathways, including *Axin2*, *β-catenin*, *Lef1*, *Hey1*, *Hes1* and *Maml1* remained unchanged between mutant and control (*Figure 4B*).

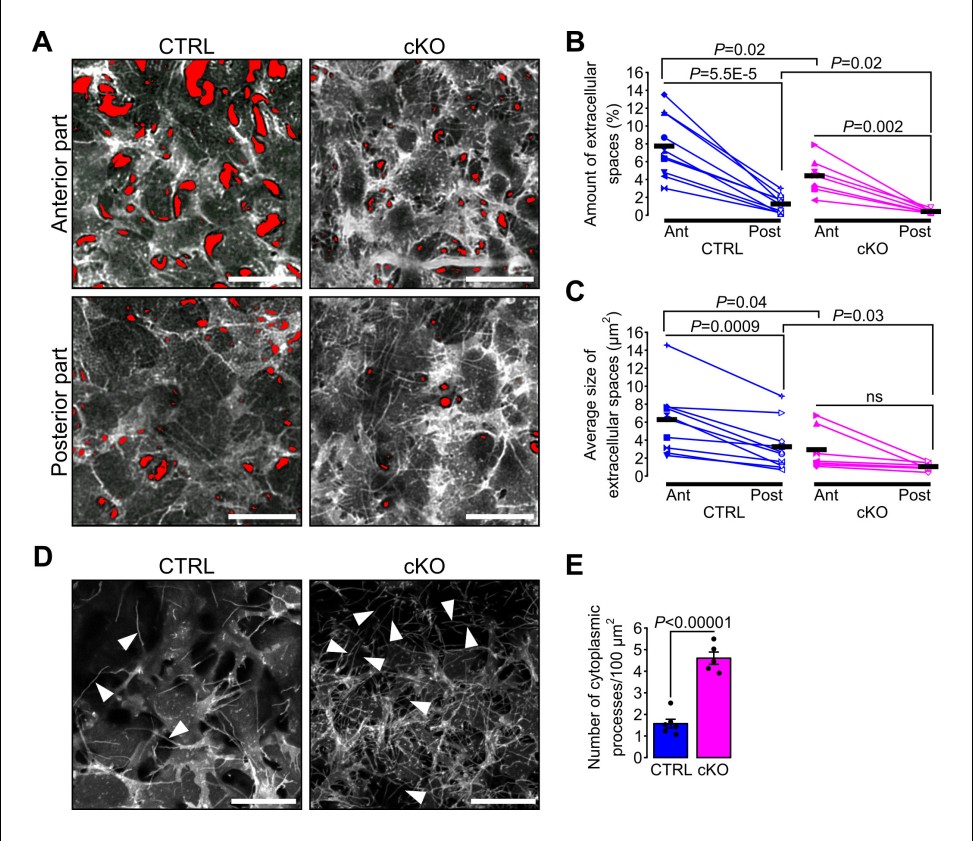

**Figure 3.** Primary cilium ablation in NCC impairs the spatial organization and the morphology of keratocytes. (**A**) Representative ex vivo confocal optical sections taken at different levels (Anterior or Posterior) of the corneal stroma of E18.5 cKO and control embryos crossed with the mT/mG reporter. Here, only the mG fluorescent signal is displayed. The red areas indicate the extracellular spaces segmented with Ilastik. Images of the anterior and posterior parts were picked from the first tenth of the stroma underlying the corneal epithelium and from the last tenth of the stroma above the corneal endothelium, respectively. Scale bar, 25 μm. (**B–C**) Each symbol represents the mean value of the amount and the average size of the extracellular space areas (in red) measured in the anterior (filled symbols) and the posterior (empty symbols) parts of the corneal stroma of a single embryo. Paired values in the graphs are connected by a solid line. Scatterplots show that the amount of extracellular spaces is significantly lower in the posterior part of the corneal stroma of both genotypes, but the average size of extracellular spaces is only significantly reduced in the posterior part of the control embryos (n = 10 control, n = 7 cKO, one symbol/embryo). Black lines represent the mean value of each group. Statistical significance was assessed using unpaired two-tailed Student's *t*-test (CTRL vs cKO), and using paired two-tailed Student's *t*-test (Ant vs Post). ns, non-significant, p≥0.05. (**D**) Representative maximum intensity projections of the first layer of stromal keratocytes underlying the corneal epithelium (only the mG reporter is displayed). Cytoplasmic processes (white arrowheads) are numerous in the cKO embryos. Scale bar, 20 μm. (**E**) Quantification of the number of cytoplasmic processes in the first layer of keratocytes underlying the corneal epithelium (n = 5–6 embryos/group). Data are presented as mean SEM. Statistical significance was assessed using two-tailed Student's *t*-test.
The online version of this article includes the following figure supplement(s) for figure 3:

**Figure supplement 1.** Segmentation of the extracellular spaces with Ilastik to study corneal stroma organization.
**Figure supplement 2.** TGFβ signaling pathway is not affected by the primary cilium ablation in NCC.

To determine the spatial distribution of Hh responsive cells in the POM during eye development we crossed cKO and control mice with the *Gli1-LacZ* reporter mouse line (*Bai et al., 2002*). In E12.5 control mice, we detected intense Hh activity in most of the POM, as all cells were strongly LacZ-positive (*Figure 4C*). At E14.5 and E18.5, the Hh activity in the POM was greatly decreased with only a subpopulation of POM cells surrounding the RPE layer remaining positive. This layer of Hh responsive cells extended from the optic nerve area to the iridocorneal angle in control embryos (*Figure 4C*). By adulthood, the Hh signaling remained active only in the choroid area, around the

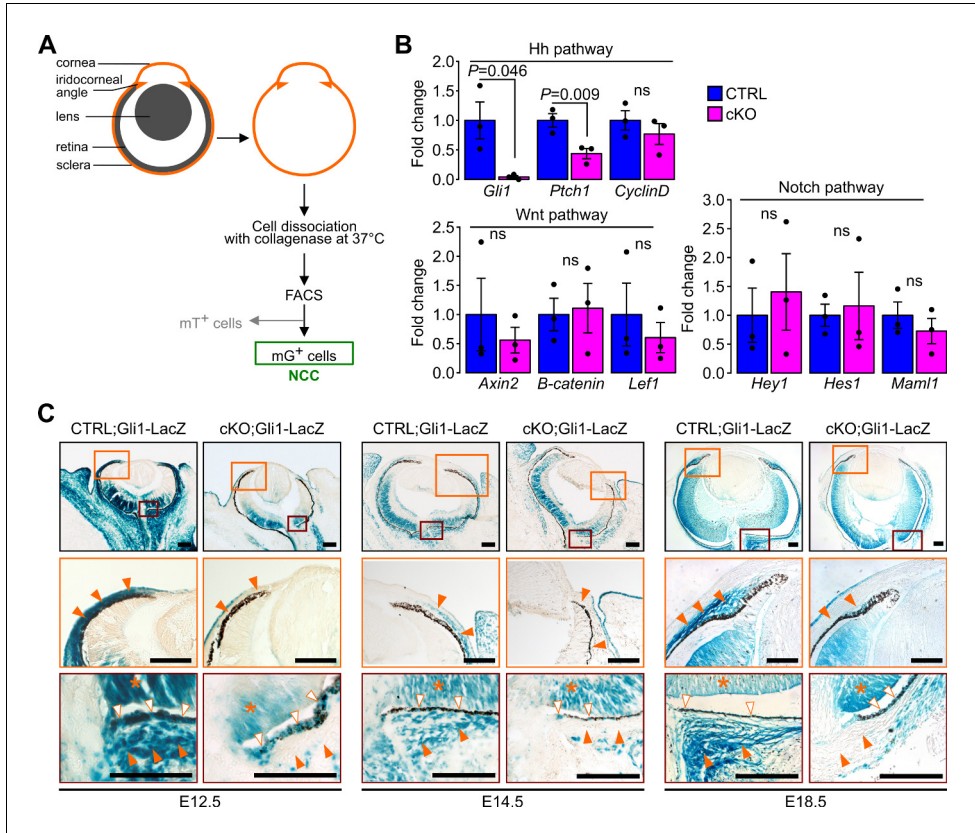

**Figure 4.** Ablation of Ift88 in NCC reduces Hh signaling in the POM NCC. (**A**) Workflow followed to collect the NCC from the eyeball by cell sorting. (**B**) Hh, Wnt and Notch target gene expression in NCC at E18.5. Fold change is expressed as mean ± SEM (n = 3 embryos/group, both eyes of each embryo were pooled together). Only the expression of Hh pathway target genes was affected by the primary cilium ablation in the NCC. Statistical significance was assessed using two-tailed Student's *t*-test. ns, non-significant, p≥0.05. (**C**) Hh activity assessed by Gli1-LacZ staining throughout the embryonic development. In the control embryos, the Hh activity progressively decreases in the POM from E12.5 to E18.5 but remains activated in a subpopulation of POM cells surrounding the RPE layer and extending until the iridocorneal angle, as well as in the POM surrounding the optic nerve (orange arrowheads). The primary cilium ablation in NCC leads to the absence of Hh activity in these specific areas. Hh signaling remains active in non-NCC derived tissues in the cKO embryos like in the RPE (white arrowheads) and the retina (asterisks). Scale bar, 100 μm.

The online version of this article includes the following figure supplement(s) for figure 4:

**Figure supplement 1.** Hh activity in the eye at adulthood.

optic nerve, and in the ciliary body (*Figure 4—figure supplement 1*). In contrast, in cKO mice, the Hh activity was nearly completely absent in the POM at any developmental stages. In both mutant and control mice, we detected intense Hh activity in tissues derived from the neuroectoderm including the retina and RPE at all developmental stages (*Figure 4C*) (*Wang et al., 2002*). Interestingly, no Hh activity was detected in the cornea of cKO or control mice between E12.5 and E18.5 despite the presence of cilia in keratocyte precursors throughout development (*Figure 4C*). Thus, ablation of the primary cilium in NCC disrupted the Hh activity in POM.

The Hh signaling pathway plays an important role in promoting cell proliferation in different tissues and cell types (*Jia et al., 2015*). We therefore hypothesized that Hh activity could influence cell proliferation rates in the Hh-active POM subpopulation surrounding the RPE cell layer. In E14.5 embryos, Hh activity in the POM was restricted to a 20 μm thick cell layer (0–20 μm) as measured from the RPE cell layer in sagittal sections of the eye (*Figure 5A*). In addition, we also arbitrarily defined a Hh-negative area of the POM encompassed between 20 μm and 50 μm from the RPE cell layer functioning as an internal control together with the Hh-negative cornea. Proliferation rates

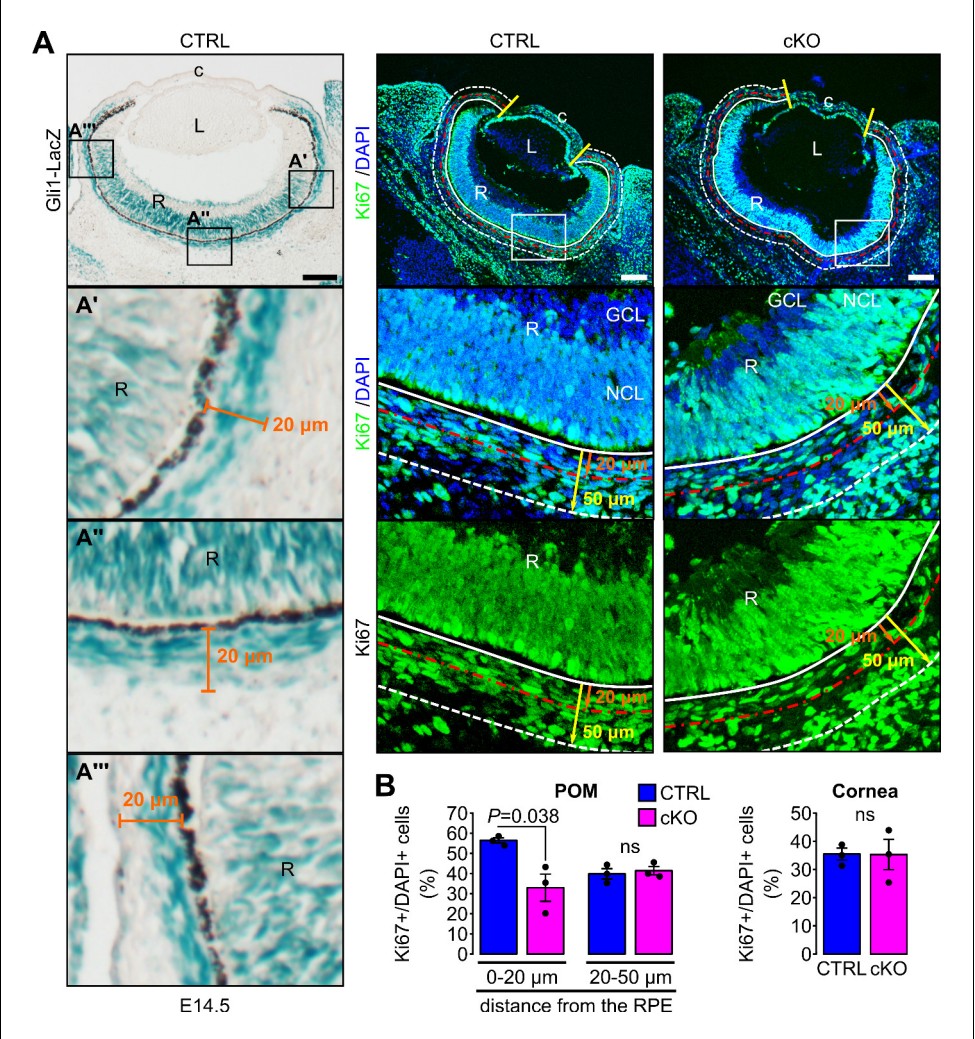

**Figure 5.** Primary cilium ablation in NCC decreases the cell proliferation specifically in the POM cells surrounding the RPE layer. (**A**) Cell proliferation was assessed at E14.5 in the cornea (delimited by the two yellow lines) and in two areas of the POM. The first area (0–20 μm) extended 20 μm from the white solid line, indicating the RPE cell layer (used as reference at 0 μm), to the red line (at 20 μm). This area corresponded to the mean thickness of Gli1LacZ-positive POM cells surrounding the RPE layer. The second area (20–50 μm) extending 30 μm from the red line to the white dashed line positioned at 50 μm from the RPE layer (white solid line). Scale bar, 100 μm; c, cornea; GCL, ganglion cell layer; L, lens; NCL, neuroblastic cell layer; R, retina. (**B**) Percentage of Ki67+ cells in the different areas of the POM and in the cornea. The number of Ki67+/DAPI+ cells is expressed as mean ± SEM. Statistical significance was assessed using one-tailed Student's $t$-test (n = 3 embryos/group). ns, non-significant, p≥0.05.

The online version of this article includes the following figure supplement(s) for figure 5:

**Figure supplement 1.** Cell proliferation assessed by BrdU pulse labeling.

were assessed by counting the number of cells stained with an anti-Ki67 or anti-BrdU Ab 2 hr after BrdU injection and normalized to the total number of cells in the same area stained with DAPI or to the total area, respectively. At E14.5, cell proliferation was significantly decreased in the POM close to the RPE layer (0–20 μm) in the cKO mutants compared to control (*Figure 5B*, *Figure 5—figure supplement 1*). In contrast, proliferation rates remained unchanged in the outer region of the POM (20–50 μm) and in the cornea where Hh was normally not activated (*Figure 5B*, *Figure 5—figure supplement 1*). Thus, ablation of the primary cilium in NCC led to a decreased cell proliferation specifically in a subpopulation of POM cell surrounding the RPE due to local inactivation of the Hh signaling.

## The choroidal secreted Ihh is the predominant ligand in the POM during development

Next, we sought to determine how Hh signaling is maintained in a restricted subpopulation of NCC within the POM. Because mammals express three different Hh homologues, *Dhh*, *Ihh* and *Shh*, we analyzed dissected tissue of the eye from E18.5 mice, including NCC, by RT-qPCR to determine which Hh ligand is expressed, as shown in the scheme in *Figure 6A*. We observed that expression of *Ihh* was significantly higher than *Dhh* and *Shh* in the sclera including the RPE layer, but not in the cornea (*Figure 6A*). As expected, we found that *Shh* but not *Dhh* or *Ihh* was expressed in the isolated neural retina where NCC are excluded (*Figure 6A*).

To determine the source of the Hh ligand we sorted different cell types of the sclera and analyzed by RT-qPCR. A previous study showed by in situ hybridization and immunohistochemistry an overlapping staining between *Ihh* and collagen type IV, a vascular endothelial marker, suggesting a possible involvement of the choroid endothelium in *Ihh* production (*Dakubo et al., 2008*). Thus, we isolated by FACS choroidal endothelial cells using the mT/mG fluorescent reporter combined to an anti-CD31 Ab (a specific marker of endothelial cells). Three different cell populations were sorted: choroidal endothelial cells (CD31$^+$, mT$^+$, mG$^-$), NCC (CD31$^-$, mT$^{low}$, mG$^+$) and cells which were neither NCC nor choroidal endothelial cells (CD31$^-$, mT$^+$, mG$^-$; *Figure 6B*). At E18.5, among the non-NCC, *Ihh* was significantly more expressed by the choroidal endothelial cells than *Dhh* and *Shh* (*Figure 6C*). Thus, we provided molecular evidence that endothelial cells of the POM produce the Hh ligand *Ihh* in the sclera.

As shown in *Figure 4*, during eye development NCC of the peripheral POM progressively lost the Hh signaling activity while Hh-responsive cells remained confined to the choroid area and the iridocorneal angle. To gain mechanistic understanding of how this process occurs we analyzed the presence of cilia and Hh components in the ciliary compartment of Hh-negative POM cells. In the presence of a Hh ligand, its receptor PTCH exits the ciliary compartment allowing SMO to concentrate in the cilium, an essential step to activate Hh signaling (*Elliott and Brugmann, 2019*). To determine whether POM cells were still ciliated and able to respond to a Hh signal, we stained primary cilia and SMO of wild-type sclera in whole mount preparations. Confocal optical sections of the POM between the RPE layer and the periphery of the sclera were collected as indicated in *Figure 6D*. At E14.5 and P0, all cells of the POM were ciliated (*Figure 6E*) and therefore we excluded the possibility that NCC of the peripheral POM lost Hh activity due to resorption of the primary cilium. At E14.5, we observed an accumulation of SMO in primary cilia on the inner cell layers of the POM surrounding the RPE (0–18 μm), consistent with the Hh activity detected in these cell layers (*Figure 4*). In contrast, we did not detect SMO in the primary cilium of cells of the peripheral POM (>18 μm from the RPE) (*Figure 6E–F*). At P0, SMO was undetectable in primary cilia of NCC in the entire POM (*Figure 6E–F*). However, when eyeballs from P0 mice were treated with a SMO agonist (SAG), SMO accumulated in cilia of POM cells (*Figure 6E–F*). This implies that POM cells of the P0 sclera were still able to activate Hh signaling upon ligand stimulation, suggesting that the decreased Hh activity in the POM during ocular development is controlled by the diffusion of the Hh ligand *Ihh* in the POM.

## Primary cilium ablation in NCC leads to corneal neovascularization

Heterozygote mutations in *FOXC1* and *PITX2* genes account for ~40% of ASD cases (*D'haene et al., 2011*). In mice, *Foxc1* and *Pitx2* haploinsufficiency leads to ocular phenotypes recapitulating human conditions of ASD including iridocorneal angle abnormalities, thinning and abnormal vascularization of the cornea (*Asai-Coakwell et al., 2006*; *Gage et al., 2014*; *Kidson et al., 1999*; *Seo et al., 2012*). Because these phenotypes were also detected in the cKO mouse, we tested whether absence of cilia in the NCC affected the expression of ASD genes in E18.5 cKO embryonic corneas. RT-qPCR analysis revealed that expression of *Foxc1* and *Pitx2* was significantly decreased in corneas of cKO embryos compared to controls, while *Pax6* expression remained unchanged (*Figure 7A*). Because *FOXC1* and *PITX2* are indispensable to specify corneal angiogenic privilege (*Gage et al., 2014*; *Kidson et al., 1999*; *Reis and Semina, 2011*; *Seo et al., 2012*), we examined the neovascularization process in the cKO mutant. To visualize blood vessels, we stained whole corneas with an Ab directed to endomucin, a marker of the vascular endothelium. Abnormal spread of blood vessels was detected into the corneas of E18.5 cKO mutant embryos (*Figure 7B–C*) where vessels covered about

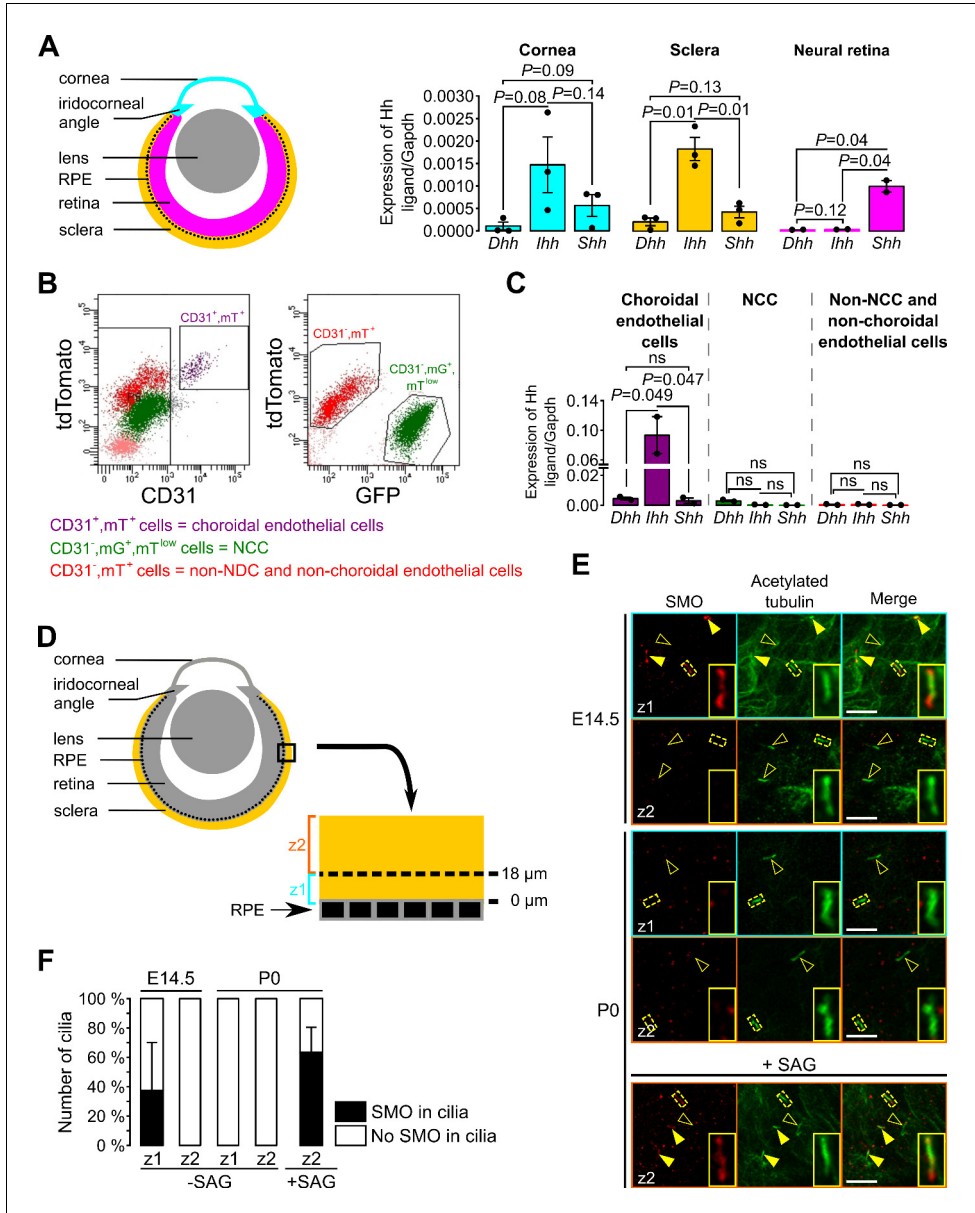

**Figure 6.** Ihh produced by choroidal endothelial cells is the predominant Hh ligand present in the POM during embryonic development. (A) Expression of *Dhh*, *Ihh* and *Shh* in the cornea (including the iridocorneal angle), the neural retina and the posterior half of the eyeball without the neural retina (corresponding to the sclera, choroid and RPE layer, here indicated as the sclera) in control embryos at E18.5 (n = 2–3). The different colors correspond to the different dissected parts of the eyeball as shown on the schematic representation on the left. (B) Representative bivariate dot plots of isolated cells from E18.5 embryonic eyeballs gated on mT and CD31 or mG expression. (C) Expression of *Dhh*, *Ihh,* and *Shh* in sorted cells isolated from E18.5 embryonic eyeballs (N = 2 independent experiments; n = 4–6 eyeballs/N). Results are mean ± SEM. Statistical significance was assessed using one-way ANOVA with posthoc Tukey HSD test. ns, non-significant, p≥0.05. (D) Schematic representation of the areas imaged in the sclera by confocal microscopy. On E14.5 embryonic sclera, we identified two different layers: z1 from 0 and 18 μm from the RPE in which SMO is accumulated in primary cilia, and z2 above 18 μm from the RPE in which SMO is not accumulated in primary cilia. For further quantifications at P0, the same layers were considered. (E) Representative images of SMO and acetylated tubulin (primary cilia) staining extracted from z1 and z2 z-stacks in whole-mount scleras at E14.5 and P0. Primary cilia with SMO accumulation are indicated with yellow arrowheads, whereas primary cilia without SMO are indicated by empty arrowheads. At P0, SMO is visible in primary cilia only upon SAG stimulation. Scale bar, 5 μm. (F) Quantification of the number of primary cilia in which SMO is present or absent (n = 5 at E14.5, n = 3 at P0).

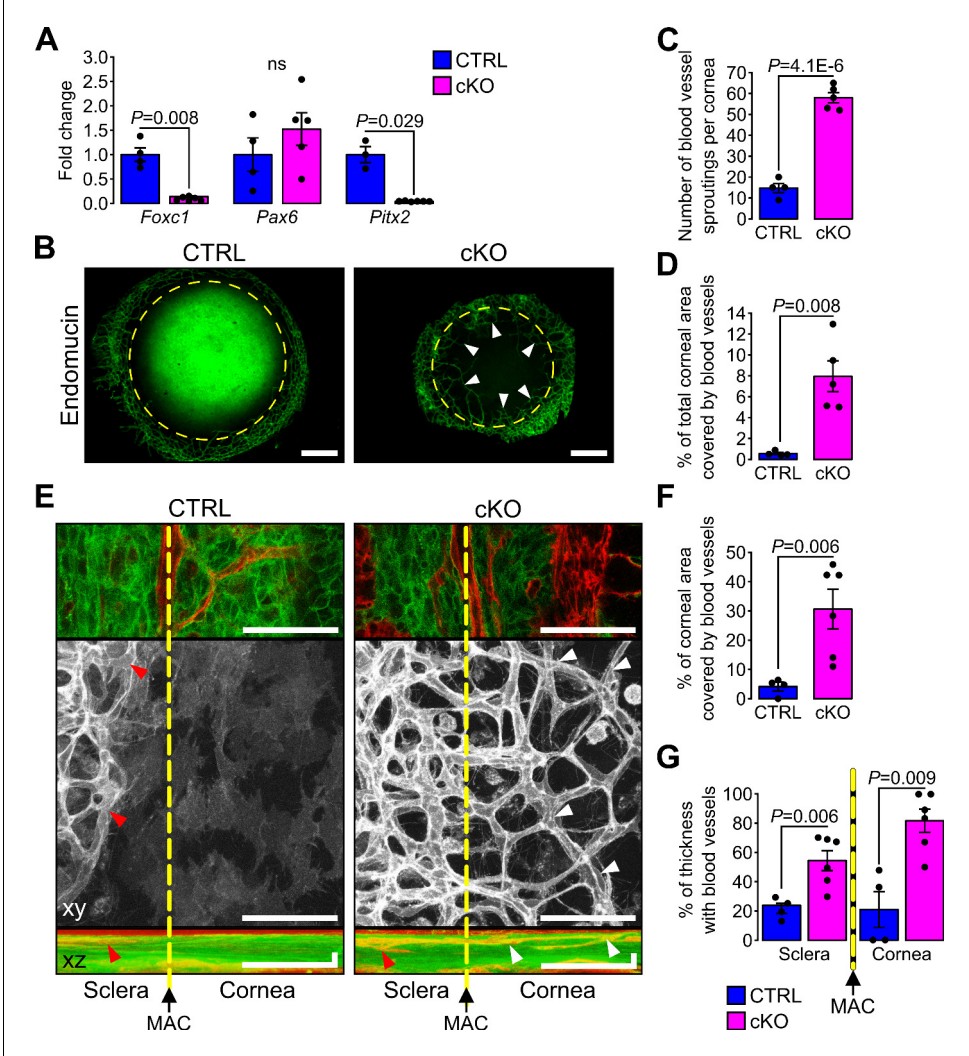

**Figure 7.** Primary cilium ablation in NCC leads to corneal neovascularization. (**A**) *Foxc1*, *Pax6* and *Pitx2* expression in the cornea (including the iridocorneal angle) (n = 3–6 embryos/group). (**B**) Representative images of whole-mount corneas stained with an anti-endomucin Ab in control and cKO embryos at E18.5. Blood vessels are indicated by white arrowheads and the yellow dotted line surrounds the cornea. Scale bar, 250 μm. (**C–D**) Quantification of the number of blood vessel sproutings and the total corneal area covered by blood vessels in E18.5 control and cKO embryos (n = 4–5 embryos/group). We defined the vessel point crossing the corneal border and extending forward to the cornea as vessel sprouting. (**E**) Representative confocal images of the cornea periphery at E18.5. The yellow dotted line represents the corneal boundary, established by the position of the major arterial circle (MAC, picture on the top). Only the mT reporter is displayed on the xy maximum intensity projection (grayscale picture), whereas both mT and mG reporters are displayed on the xz maximum intensity projection. Arrowheads indicate blood vessels which are only visible in the sclera side in controls (red arrowheads) whereas they extend into the cornea in the cKO embryos (white arrowheads). Scale bar, 25 μm. (**F**) Percentage of the corneal area covered by blood vessels at the periphery of the cornea (n = 4–6 embryos/group). (**G**) Quantification of the corneal and stroma thickness in which blood vessels are present (n = 4–6 embryos/group). Results are mean ± SD. Statistical significance was assessed using two-tailed Student's *t*-test. ns, non-significant, p≥0.05.

The online version of this article includes the following figure supplement(s) for figure 7:

**Figure supplement 1.** Live imaging of the cornea boundary area.

8% of the total corneal area (*Figure 7D*). To gain additional insight into the abnormal neovascularization process, we performed live imaging using confocal microscopy at the cornea periphery using enucleated eyes from E18.5 embryos with the *mT/mG* reporter. Due to their mesenchymal origin, blood vessels were easily detectable as red (mT) tubular networks clearly distinct from the green (mG) NCC-derived POM (*Figure 7—figure supplement 1*). By confocal optical sectioning we identified the major arterial circle which served as a reproducible reference between the end of the sclera and the beginning of the cornea. The average area occupied by blood vessels in microscope fields selected at the corneal periphery occupied ~30% of the total area in the cKO but only ~4% in the control (*Figure 7E–F*). In addition, while peripheral blood vessels of the control were present only in superficial layers of the sclera-cornea interface on both sides of the major arterial circle (~20% of the thickness), those of the mutant invaded the lower layers of both, the sclera (~50% of the thickness) and the cornea (~80% of the thickness). This demonstrates that ablation of the primary cilium in the NCC leads to the loss of the angiogenic privilege in the cornea as well as vascular abnormalities in the sclera, potentially due to the lowered expression of *Foxc1* and *Pitx2*.

## Primary cilium ablation in NCC impairs early corneal innervation and centripetal migration of the sensory nerves

Concurrent with the establishment of the angiogenic privilege is corneal innervation (*Lwigale, 2015*). Because corneal sensory nerves are derived from the neural crest part of the trigeminal ganglion (*Lwigale, 2001*), we assessed whether primary cilium ablation in NCC would also impact corneal innervation. During development, corneas of both control and cKO embryos were innervated (*Figure 8A*), however, at E13.5, the number of nerve bundles detected in the cKO eye was significantly lower than that of the control (*Figure 8B*). Moreover, abnormally long nerve projections across the cornea were observed in cKO embryos while nerves were only visible at the periphery of the control cornea (*Figure 8A*). At E18.5, corneas of both genotypes were fully innervated (*Figure 8A*), including the corneal epithelium (*Figure 8—figure supplement 1*). However, the corneal nerve density was significantly higher in the cKO embryos (*Figure 8C*) and the nerves in cKO corneas were less organized centripetally compared to the control (*Figure 8D–E*). The average angle θ formed by the major nerve branches with the radius of the cornea was significantly larger in cKO embryos compared to controls (*Figure 8D–E*). These data suggest that primary cilium ablation in the NCC impaired the early corneal innervation and the centripetal migration of the sensory nerves.

## Discussion

Ocular conditions such as microphthalmos/anophthalmos, aniridia, sclerocornea, abnormal corneal thickness, corneal neovascularization, abnormal iridocorneal angle, corneal opacity, glaucoma, and cataract, have been reported in patients affected by ciliopathies including Joubert and Mekel syndrome, suggesting a role of primary cilium in AS development (*Abuhamda and Elsous, 2018*; *Cortés et al., 2016*; *Eibschitz-Tsimhoni, 2003*; *Luo et al., 2014*; *MacRae et al., 1972*; *Zaki et al., 2011*). However, the pathogenesis of these conditions remains largely unknown. In the present study, we demonstrated that conditional ablation of the primary cilium in NCC led to ASD with conditions similar to those observed in humans. We showed that primary cilia are present in virtually all neural crest-derived cells of the POM including keratocytes. However, we found that primary cilia were required for the propagation of the Hh pathway in a subpopulation of POM cells surrounding the RPE layer but not in neural crest-derived cells of the cornea where the Hh pathway is not active. We showed that the expression of genes implicated in the pathogenesis of ASD, including *Foxc1* and *Pitx2*, was strongly reduced in the cornea and iridocorneal angle tissue lacking cilia in the NCC which is consistent with AS phenotypes, including the loss of corneal angiogenic privilege. Thus, we have established an important functional connection between cilia-dependent signaling and ASD. Overall, this study demonstrated the novel and fundamental role of the primary cilium in the development of structures of the AS derived from the NCC.

Mice carrying somatic null mutations in genes encoding proteins required for cilia assembly or ciliary function display severe microphthalmia and anophthalmia (*Andreu-Cervera et al., 2019*; *Burnett et al., 2017*; *Caspary et al., 2007*; *Cela et al., 2018*; *Dyson et al., 2017*; *Jacoby et al., 2009*; *Wang et al., 2018*). These conditions are likely due to aberrant Hh activity in cells of the neuroepithelium of the developing optic vesicle that precedes the development of the AS

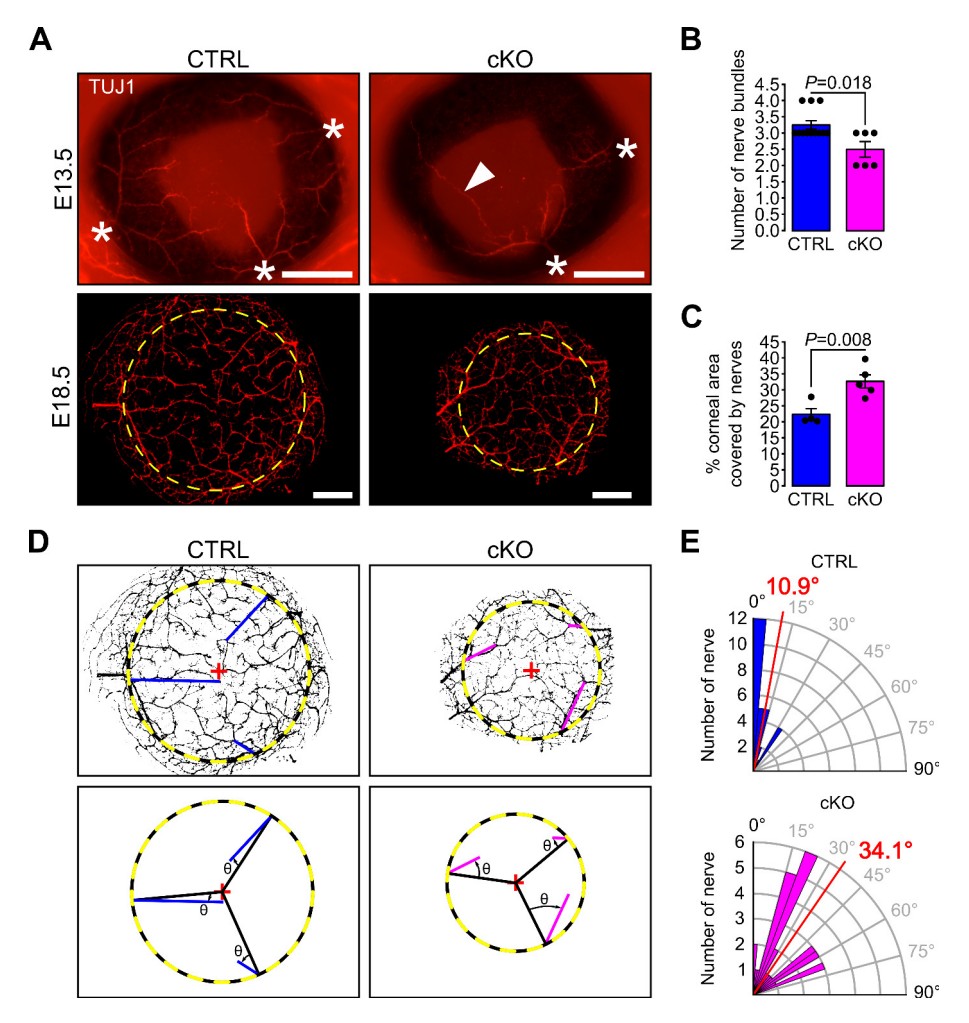

**Figure 8.** Primary cilium ablation in NCC reduces early corneal innervation and impairs the centripetal migration of sensory nerves. (**A**) Representative images of whole-mount corneas stained with an anti-TUJ1 Ab in control and cKO embryos at E13.5 and E18.5. At E13.5, three to four nerves bundles (asterisk) are present in controls and start to project at the periphery of the cornea. Abnormal nerve projections (head arrow) across the cornea are visible in some cKO embryos at E13.5. By E18.5, the cornea (surrounded by the yellow dotted line) is fully innervated in both the control and cKO embryos. Scale bar, 250 µm. (**B**) Quantification of the number of nerve bundles at E13.5 which is significantly decreased in cKO embryos compared to control (n = 8–13 embryos/group). (**C**) Nerve density at E18.5 quantified by measuring the percentage of corneal area covered by nerves (n = 4–5 embryos/group). (**D**) Corneal nerve segmentation from E18.5 embryos. The colored lines (blue in control, pink in cKO) represent the axis of major nerve branches formed between the point where the main nerve branch crosses the corneal border (yellow dotted line) and its end point (final point of the nerve branch or the branching point from which two secondary branches with similar diameter arise). The angle θ formed by the major nerve branches and the radius of the cornea (black line) was measured with Fiji. (**E**) Quantification of the angle θ in control and cKO mice at E18.5. The nerves in the cKO embryos are less centripetally organized than in control (p=8.43E$^{-7}$, n = 4–5 embryos/group). In B and C, results are mean ± SEM. In D, the mean angle is indicated in red. Statistical significance was assessed using two-tailed Student's *t*-test.

The online version of this article includes the following figure supplement(s) for figure 8:

**Figure supplement 1.** Innervation of the corneal epithelium.

---

(*Bakrania et al., 2010*). To overcome this obstacle and test the function of the primary cilium in AS development, we ablated IFT88 in the NCC, a multipotent progenitor cell population that produces a large range of differentiation fates including a large portion of AS structures (*Gage et al., 2005*).

Relevant to our study, conditional deletions of ASD causative genes such as *Foxc1*, *Pitx2* or *Ext1* in NCC, lead to defects of the AS including anophthalmos, abnormal corneal thickness, reduced anterior chamber, defective iridocorneal angle, and corneal neovascularization similar to abnormalities observed in ASD patients carrying dominant mutations in these genes (*Evans and Gage, 2005*; *Iwao et al., 2009*; *Seo et al., 2012*). While the cKO mice revealed a direct connection between the primary cilium and ASD, severe craniofacial defects caused late embryonic death and prevented us from following the postnatal development of structures such as the iris, trabecular meshwork and Schlemm's canal (*Gould et al., 2004*). Thus, further studies involving conditional ablation of the cilium in specific AS tissues will be required, notably using the Keratocan-rtTA mouse (*Zhang et al., 2017*).

Low expression levels of *Foxc1* and *Pitx2* in the cornea of the cKO mutant could explain some of its AS phenotypes, including abnormal corneal vascularization and thinning of the cornea (*Asai-Coakwell et al., 2006*; *D'haene et al., 2011*; *Evans and Gage, 2005*; *Gage et al., 2014*; *Kidson et al., 1999*; *Kume et al., 1998*; *Reis and Semina, 2011*; *Seo et al., 2012*). Mice carrying only one copy of *Foxc1* (*NC-Foxc1*$^{+/-}$) in the NCC do not lose cornea angiogenic privilege meaning their cornea remains avascular. Whereas loss of both *Foxc1* copies in the NCC leads to severe neovascularization of the whole cornea suggesting a strong relationship between the maintenance of the corneal angiogenic privilege and the expression level of *Foxc1* (*Seo et al., 2012*). Consistent with this possibility, we observed a partial vascularization in the cornea of the cKO mouse. The expression level of *Foxc1* and *Pitx2* also differentially regulates corneal thickness. While the corneal thickness increases in *NC-Foxc1*$^{-/-}$ mice and *Pitx2*$^{-/-}$, it remains unaltered in *NC-Foxc1*$^{+/-}$ and decreases in *Pitx2*$^{+/-}$ mice (*Asai-Coakwell et al., 2006*; *Gage et al., 1999*; *Seo et al., 2012*). Thus, the reduction of *Pitx2* expression could be one of the concomitant causes leading to thinning of the cornea observed in the cKO embryos. In summary, the partial corneal vascularization and decreased stroma thickness observed in the cKO embryos are consistent with a decreased expression of *Foxc1* and *Pitx2* in the cornea.

The abnormal corneal neovascularization observed in the cKO could also derive from a reduction or absence of Hh activity in the POM. In a zebrafish model, the loss of Hh signaling induces excess sprouting of blood vessels in the dorsal eye and impairs growth of blood vessels in the ventral eye (*Weiss et al., 2017*), observations which are consistent with the aberrant expansion of blood vessel domains deep into the sclera of cKO embryos (*Figure 7*). A thorough analysis of the choroid in NCC conditional mutants with reduced Hh activity obtained independently from the ablation of the primary cilium could elucidate a specific role of the Hh pathway in development and patterning of the choroid.

Decreased expression of *Foxc1* and *Pitx2* in tissues of the AS in the cKO mouse suggests a possible direct or indirect mechanism of gene expression regulation of these transcription factors by cilia-dependent signaling. A recent study showed that mutations in the *FOXC1* gene in humans cause, in addition to ASD, systemic conditions similar to those found in ciliopathic patients including polydactyly (*Lehmann et al., 2019*). This study also showed that lowering the level of *Foxc1* expression in epithelial cells induced a reduction of cilia length and interfered with cilia localization of Hh components, most notably GLI2 (*Lehmann et al., 2019*). FOXC1 also appeared to directly interact with GLI2 and promoted the expression of Ihh target genes, including *Gli1* and *Ptch1* (*Yoshida et al., 2015*). Together, these data suggest a strong relation between primary cilia, *Foxc1* expression and Hh activation, explaining why similar ocular and systemic phenotypes are observed in ASD and ciliopathic patients.

The process of corneal innervation occurs in parallel with the establishment of the corneal angiogenic privilege. The majority of corneal nerves derive from the ophthalmic division of the trigeminal ganglion of NCC origin (*Lwigale, 2001*). The cKO mouse displays a defective corneal innervation consisting of abnormalities in the nerve projections similar to those described in *Semaphorin three* deficient mouse models (*McKenna et al., 2012*; *Taniguchi et al., 1997*). *Semaphorin three* regulates the navigation process of the nerves by guiding the direction of axon growth (*Taniguchi et al., 1997*). In our ciliary model, the decreased number of nerve bundles at E13.5 and the less centripetal migration of the corneal nerves at E18.5 suggest that the navigation process of trigeminal nerves in the cornea is affected. Consistent with these exciting possibilities, ciliogenic proteins including KIF3A, IFT88, and ARL13B, are required for the directional migration of GABAergic neurons (*Baudoin et al., 2012*; *Higginbotham et al., 2012*). Thus, assembly of the primary cilium on

trigeminal nerves could be critical in reestablishing corneal innervation following corneal transplant or during recovery from corneal trauma.

The Hh signal transduction pathway is intimately linked to the primary cilium (**Bangs and Anderson, 2017**; **Reiter and Leroux, 2017**), and its activation level is critical to ensure normal eye development (**Burnett et al., 2017**; **Chiang et al., 1996**; **Gordon et al., 2018**). In this study, we showed that during early stages of AS development, virtually all POM cells are actively responding to Hh signaling. However, as the AS development progresses only a subpopulation of NCC of the POM surrounding the RPE cell layer and in the iridocorneal angle, maintains Hh responsiveness by actively expressing *Gli1* and localizing SMO to the primary cilium compartment. Moreover, the NCC derived components of the cornea, which includes keratocyte precursors and corneal endothelium, were not found to be Hh responsive at any developmental stage. It was therefore imperative to determine how differential distribution of Hh responsiveness was achieved in cells of similar embryonic origin. We found that cells of the POM maintained the ability to respond to Hh stimulation at several levels: all cells of the POM and the corneal stroma were found ciliated throughout development and after birth. In addition, in vitro SAG treatment of ocular tissues indicated that all POM cells maintained the ability to respond to Hh stimuli by promoting SMO localization to their primary cilium. We identified CD31 positive cells of the POM as the main source of Ihh, suggesting that endothelial cells of the choroid vasculature could be the principal source of Ihh in the eye. Consistent with this possibility, a previous study showed that *Ihh* expression in the POM colocalizes with collagen IV, a marker for the vasculature, next to the RPE layer (**Dakubo et al., 2008**). Thus, the secretion of Ihh by the choroid vessels is likely to regulate the propagation of the Hh responsiveness in the periocular space and the cornea.

Our data suggest that one of the main functions of Hh activity in the POM during late gestation is to promote cell division in the subpopulation of NCC surrounding the RPE layer. Surprisingly, although the size of the cornea in the cKO mouse was considerably reduced, the number of keratocyte precursors within the corneal stroma was similar to that of control corneas. Thus, the reduction of cell proliferation in this subpopulation of POM cells did not account for the number of cells that populate the cornea. However, we identified a group of mesenchymal cells at the iridocorneal angle in control mice that were absent or reduced in number in the cKO mutant. These cells could be precursors of drainage structures of the AS, such as the trabecular meshwork and/or the iris stroma. Comparative studies involving differential single cell analysis of the POM cells from the AS of the cKO and control mice will therefore be crucial to identify this distinct cell population at the iridocorneal angle.

In conclusion, we have shown that primary cilia are indispensable for normal development of NCC-derived structures in the ocular AS. When the primary cilium is ablated in NCC, an ASD phenotype is induced including small and thinner cornea, abnormal iridocorneal angle, disorganized corneal stroma and neovascularization. In addition, primary cilium ablation in NCC reduces the cell proliferation in a subpopulation of POM cells surrounding the RPE which are normally responsive to Ihh, the main Hh ligand expressed by endothelial cells of the choroid. Moreover, primary cilium ablation in NCC decreases the expression of transcription factors associated with ASD in patients. These findings suggest that the primary cilium might contribute to the pathology of ASD, and associated complications like glaucoma.

## Materials and methods

**Key resources table**

| Reagent type (species) or resource | Designation | Source or reference | Identifiers | Additional information |
|---|---|---|---|---|
| Strain, strain background (*Mus musculus*) | *Ift88(tm1Bky)* | (**Haycraft et al., 2007**), | RRID: MGI:6315331 | |
| Strain, strain background (*Mus musculus*) | Wnt1-Cre | (**Danielian et al., 1998**), | RRID: IMSR_JAX:009107 | |

*Continued on next page*

Continued

| Reagent type (species) or resource | Designation | Source or reference | Identifiers | Additional information |
|---|---|---|---|---|
| Strain, strain background (*Mus musculus*) | B6.129(Cg)-Gt(Rosa)26Sor (tm4(ACTB-td Tomato,-EGFP)Luo)/J | (*Muzumdar et al., 2007*) | Jackson laboratories stock No 007676 RRID: IMSR_JAX:007676 | |
| Strain, strain background (*Mus musculus*) | Gli1(tm2Alj)/J | (*Bai et al., 2002*) | Jackson Laboratories stock No 008211 RRID: IMSR_JAX:008211 | |
| Strain, strain background (*Mus musculus*) | Gt(ROSA)26Sor (tm1(Sstr3/GFP)Bky) | (*O'Connor et al., 2013*) | RRID: ISMR_JAX:024540 | |
| Antibody | Anti-acetylated tubulin (mouse monoclonal) | Sigma-Aldrich | Cat #: 6-11B-1 RRID: AB_527348 | 1:1 |
| Antibody | Anti-Arl13b (rabbit polyclonal) | ProteinTech Group | Cat #: 17711–1-AP RRID: AB_2060867 | 1:800 |
| Antibody | Anti-BrdU (rat monoclonal) | Abcam | Cat #: ab6326 RRID: AB_305426 | 1:400 |
| Antibody | Anti-CD31 (rat monoclonal) | BD Biosciences | Cat #: 550274 RRID: AB_393571 | 1:500 |
| Antibody | Anti-endomucin (rat monoclonal) | eBioscience | Cat #: 14-5851-85 RRID: AB_891531 | 1:500 |
| Antibody | Anti-Ki67 (rabbit polyclonal) | Abcam | Cat #: ab15580 RRID: AB_443209 | 1:300 |
| Antibody | Anti-pSmad2/3 (goat polyclonal) | Santa Cruz | Cat #: sc11769 RRID: AB_2193189 | 1:200 |
| Antibody | Anti-SMO (rabbit polyclonal) | MilliporeSigma | Cat #: ABS1001 | 1:100 |
| Antibody | Anti-TUJ1 (rabbit monoclonal) | Covance | Cat #: MRB-435P RRID: AB_663339 | 1:500 |
| Antibody | Alexa Fluor 594 goat anti-rabbit IgG | Invitrogen | Cat #: A27034 RRID: AB_2536097 | 1:200 |
| Antibody | Alexa Fluor 647 donkey anti-rat | Jackson ImmunoResearch | Cat #: 712-605-153 RRID: AB_2340694 | 1:200 |
| Antibody | FITC donkey anti-mouse | Jackson ImmunoResearch | Cat #: 715-096-151 RRID: AB_2340796 | 1:200 |
| Antibody | FITC donkey anti-rat | Jackson ImmunoResearch | Cat #: 712-097-003 RRID: AB_2340655 | 1:200 |
| Antibody | FITC goat anti-rat IgG2a | Bethyl | Cat #: A110-109F RRID: AB_67287 | 1:200 |
| Antibody | TRITC donkey anti-goat | Jackson ImmunoResearch | Cat #: 705-025-147 RRID: AB_2340389 | 1:200 |
| Antibody | TRITC donkey anti-rabbit | Jackson ImmunoResearch | Cat #: 711-025-152 RRID:AB_2340588 | 1:200 |
| Chemical compound, drug | BrdU | Sigma-Aldrich | Cat #: B5002 | 10 mg/mL |
| Chemical compound, drug | Collagenase | Sigma | Cat #: C2674 | 8 mg/mL |

*Continued*

| Reagent type (species) or resource | Designation | Source or reference | Identifiers | Additional information |
|---|---|---|---|---|
| Chemical compound, drug | SMO agonist (SAG) | Calbiochem | Cat #: 566660 | 100 nM |
| Chemical compound, drug | X-gal | LabScientific | Cat #: X266 | |
| Commercial assay or kit | RNeasy microkit | Qiagen | Cat #: 74004 | |
| Software, algorithm | Fiji | (*Schindelin et al., 2012*) | RRID: SCR_002285 | |
| Software, algorithm | Ilastik | (*Haubold et al., 2016*) | RRID: SCR_015246 | |

## Mouse strains

Mouse strains *Ift88^{tm1Bky}*, here referred to as *Ift88^{fl/fl}* (*Haycraft et al., 2007*), *Wnt1-Cre* (*Danielian et al., 1998*), *B6.129(Cg)-Gt(Rosa)26Sor^{tm4(ACTB-tdTomato,-EGFP)Luo}/J* (*R26^{mT/mG}*, Jackson Laboratories stock No 007676) (*Muzumdar et al., 2007*), and *Gli1^{tm2Alj}/J* (*Gli1-LacZ*, Jackson Laboratories stock No 008211) (*Bai et al., 2002*) were maintained on mixed C57Bl/6, FVB and 129 genetic backgrounds. *Gt(ROSA)26Sor^{tm1(Sstr3/GFP)Bky}* mouse (here referred to as Sstr3::GFP mouse) (*O'Connor et al., 2013*) was maintained on a mixed CD1 background. *Ift88* conditional knockout (cKO) were generated by crossing *Wnt1-Cre;Ift88^{fl/+}* males with *Ift88^{fl/fl}* females. Since *Wnt1-Cre; Ift88^{fl/fl}* mutants die at birth, all the studies were conducted during the embryonic development. *Wnt1-Cre;Ift88^{fl/+}* and *Ift88^{fl/fl}* littermates were used as control. For time breeding, females were examined daily for vaginal plug. The day of the plug was considered embryonic day 0.5 (E0.5). All animal procedures were performed in accordance with the guidelines and approval of the Institutional Animal Care and Use Committee at Icahn School of Medicine at Mount Sinai, at Johns Hopkins University, and at the University of Pennsylvania.

## Histology, electron microscopy and immunofluorescence

Mouse embryos at different gestational stages were removed from *Ift88^{fl/fl}* females crossed with *Wnt1-Cre;Ift88^{fl/+}* males and pictures were taken with a Leica M80 stereomicroscope equipped with a Leica color IC80 HD camera. Half head or enucleated eyes from the embryos were fixed overnight in 4% paraformaldehyde (PFA) and embedded in paraffin for histological analysis. Hematoxylin and eosin (HE) staining was performed following standard procedures. Corneal stromal thickness was determined in the center of the cornea on HE sections from three mice per age.

Endomucin and TUJ1 staining were performed on fixed whole corneas as previously described (*McKenna and Lwigale, 2011*; *McKenna et al., 2014*). For immunofluorescence (IF) and X-gal staining on sections, half head and enucleated eyes from embryos were embedded and frozen in optimal cutting temperature compound (OCT Tissue-Tek, Sakura Finetek, Torrance, CA). For detection of the β-galactosidase activity by X-Gal staining, sections were fixed 10 min with 0.2% glutaraldehyde, 2 mM $MgCl_2$ in PBS at room temperature, and then incubated overnight in a solution containing 1 mg/mL Xgal, 5 mM $K_3Fe(CN)$, 5 mM $K_4Fe(CN)$, 0.01% deoxycholate, 0.02% NP40 and 2 mM $MgCl_2$ in PBS at 4°C. Sections were imaged with a Zeiss AxioImager.Z2M microscope equipped with a Zeiss Axiocam 503 color camera. For immunofluorescent staining, sections were fixed 10 min with 4% PFA at room temperature. After blocking with 2% normal donkey serum (Jackson ImmunoResearch, West Grove, PA), 0.1% Triton in PBS for 30 min at room temperature, primary cilia were stained using a rabbit anti-Arl13b Ab (17711–1-AP, Proteintech Group, Rosemont, IL), pSmad2/3^+ cells were stained using a goat anti-pSmad2/3 Ab (sc11769, Santa Cruz, Dallas, TX) and proliferative cells were stained using a rabbit anti-Ki67 Ab (ab15580, Abcam, Cambridge, United Kingdom). Nuclei were counterstained with DAPI. Confocal pictures were acquired with a Leica DM6000.

For transmission electron microscopy (TEM), embryonic eyes were processed as previously described (*Grisanti et al., 2016*). Briefly, samples were fixed with 1% PFA and 3% glutaraldehyde in 0.1 M sodium cacodylate buffer, post-fixed with 1% osmium tetroxide, embedded in Epon (Electron

Microscopy Sciences, Hartfield, PA) and stained with uranyl acetate and lead citrate after sectioning. Sections were imaged with Hitachi H7650 or S4300 microscopes.

## BrdU pulse labeling

Pregnant females at E14.5 received a single intraperitoneal injection of 50 mg/kg body weight of 5-bromo-2-deoxyuridine (Sigma-Aldrich, Saint-Louis, MO), were sacrificed after 2 hr and embryos were collected. After fixation with 4% PFA for 15 min, half heads were embedded, frozen in OCT and cryosectioned. After 10 min of postfixation with 4% PFA and wash with PBS, sections were incubated in 2 N HCl for 25 min at 37°C and subsequentially processed for immunofluorescence using a rat anti-BrdU Ab (ab6325, Abcam) as described above.

## SAG stimulation

Enucleated eyes from P0 mice were cut in half on the sagittal axis and the lens and the retina were removed. Eyeball pieces were incubated for 5 hr in keratinocyte-SFM with or without 100 nM SMO agonist (SAG, MilliporeSigma, Burlington, MA) at 37°C in 5% CO2. Tissues were fixed 12 min with 4% PFA in PBS, incubated 12 min with 5% Triton in PBS and then primary cilia and SMO were stained using a mouse anti-acetylated tubulin Ab (6-11B-1, Sigma-Aldrich) and a rabbit anti-SMO Ab (1:100, ABS1001 MilliporeSigma), respectively. Whole-mount eyeball tissues were imaged using a Zeiss LSM800 confocal. Cilia and SMO staining were also performed on ocular tissues from E14.5 embryos, without prior incubation with or without SAG.

## Ex vivo imaging

Confocal imaging was performed on whole eyes from E18.5 *Wnt1-Cre;Ift88$^{fl/fl}$;R26$^{mT/mG}$* and *Wnt1-Cre;Ift88$^{fl/+}$;R26$^{mT/mG}$* embryos as previously described (*Grisanti et al., 2016*). Live GFP and tdTomato fluorescences were recorded by a Zeiss LSM880 confocal microscope, using a 40X water immersion objective lens and used to scan a 0.05 mm$^2$ field of view to study the keratocyte organization and 0.005 mm$^2$ field of view to quantify the cytoplasmic processes. For the keratocyte organization, serial optical sections were acquired in 0.45 µm steps, through the entire cornea, from the corneal epithelium to the lens epithelium (49–89 µm per z-stack). For the quantification of cytoplasmic processes, serial optical sections were acquired in 0.2 µm steps, from the last epithelial cell layer before the corneal stroma to the first layer of keratocytes (~5 µm per z-stack).

Quantification of the protrusions in Wnt1-Cre-positive keratocytes was done by counting cytoplasmic processes at the interface between corneal epithelium and stroma. From the acquired z-stacks, images from the first one where green fluorescent cytoplasmic processes are visible to the last one before keratinocyte cell bodies are imaged were selected and a maximum intensity projection was done with Fiji (*Schindelin et al., 2012*). For each mouse, cytoplasmic processes were manually counted in four distinct areas in the center of the cornea and expressed as number of cytoplasmic processes per 100 µm$^2$. Keratocyte organization in the stroma was assessed by quantifying the amount and the average size of extracellular spaces in the first 1/10 of the z-stack just under the corneal epithelium, corresponding to the anterior part of the stroma, and the last 1/10 of the z-stack above the corneal endothelium, corresponding to the posterior part of the stroma (2.25–4.5 µm thickness/stack). After segmentation of the stacks with Ilastik (*Haubold et al., 2016*), the amount and average size of extracellular spaces were quantified using Fiji (*Figure 3—figure supplement 1*). The amount of extracellular spaces was defined as the sum of the areas of each extracellular space (shown in red in *Figure 3A*) detected in a single confocal optical section and normalized to the total area of the section (5–12 optical sections were analyzed per embryo for each selected stack). The amount of extracellular spaces was expressed as % of the total area of the section. The average size of extracellular spaces was quantified from the same optical sections and expressed in µm$^2$.

For visualization of the blood vessels at the corneal periphery, the microscope field was centered on the major arterial circle which was considered as the border between the cornea and the sclera (*Figure 7—figure supplement 1*). Serial optical sections were acquired from the epithelium to the presumptive iris (49 to 109 µm per z-stack). After maximum intensity projections, the corneal area covered by blood vessels and the percentage of corneal and scleral thicknesses with blood vessels were quantified with Fiji.

## Intravital imaging

Intravital imaging of cilia in the mouse cornea was performed with an Olympus FV1200MPE microscope, equipped with a Chameleon Vision II Ti:Sapphire laser. Mice were anesthetized with intraperitoneal injection of ketamine and xylazine (15 mg/mL and 1 mg/mL, respectively in PBS). A custom made stage was used to immobilize the mouse head and expose the eye globe. Mice were then placed under the microscope onto a heating pad and kept anesthetized with a continuous delivery of isoflurane through a nose cone (1% in air). A laser beam tuned at 930 nm was focused through a 25X water immersion objective lens (XLPlan N, N.A.1.05; Olympus USA) and used to scan a 0.5 $mm^2$ field of view. Serial optical sections were acquired in 2–3 µm steps, starting from the surface of the eye and capturing the entire thickness of the cornea (epithelium ~40 µm, stroma ~80 µm).

## Cell sorting

*Unstained cells*. At E18.5, eyes from *Wnt1-Cre;Ift88$^{fl/+}$;R26$^{mT/mG}$* and *Wnt1-Cre;Ift88$^{fl/fl}$;R26$^{mT/mG}$* embryos were enucleated. After removing the lens and retina from the eyeball, remaining tissues were digested by 8 mg/mL collagenase (Sigma) in keratinocyte-SFM (Thermo Fischer, Waltham, MA) for 1 hr at 37°C under stirring. Isolated cells were centrifuged for 5 min at 1800 rpm and re-suspended in sorting buffer (2% FBS in PBS) for cell sorting with a BD FACSAria (BD Biosciences, San Jose, CA).

*Stained cells*. Isolated cells from E18.5 *Wnt1-Cre;Ift88$^{fl/+}$;R26$^{mT/mG}$* and *Wnt1-Cre;Ift88$^{fl/fl}$;R26$^{mT/mG}$* embryonic scleras were incubated 45 min with a rat anti-CD31 Ab (550274, BD Biosciences, 1:500), washed, and then incubated 30 min with an Alexa Fluor 647 donkey anti-rat secondary Ab. After washing, cells were re-suspended in sorting buffer for cell sorting with a BD FACSAria (BD Biosciences).

## Quantitative RT-PCR

Total RNA from the cornea dissected just under the iridocorneal angle, neural retina, the posterior half of the eyeball without the neural retina (corresponding to the sclera and the RPE layer) or sorted cells were extracted using RNeasy microkit (Qiagen, Hilden, Germany) according manufacturer instructions. One hundred nanogram of RNA were reverse transcribed as previously described (*Grisanti et al., 2016*). Real-time PCR was performed in triplicate using Maxima SYBR Green Master Mix (Thermo Fisher) on ABI PRISM 7900HT (Applied Biosystems, Foster City, CA). PCR program consisted of 40 cycles of 95°C for 15 s, 55°C for 15 s and 72°C for 30 s. Data were analyzed using the 2-Δ Ct method, with *Gapdh* transcript as a reference. Primer sequences are listed in table in *Supplementary file 1*.

## Statistical analysis

Data are presented as mean ± SEM. Student's *t*-tests were performed with Excel 2017 (Microsoft) and one-way ANOVA with post-hoc Tukey HSD test was performed with the online web statistical calculators https://astatsa.com/. A *P* value < 0.05 was considered significant.

## Acknowledgements

We thank L Grisanti and Q Liu for their involvement during the early stages of this project, members of the Mlodzik and Iomini labs as well as members of the Wilmer Cornea Group for helpful inputs and discussions. We thank Nada Marjanovic and Pei-Yu Kuo for their help with qPCR and cell sorting experiments. We are particularly grateful to Rebecca Sherburn and James Foster for critical comments to the manuscript. This work was supported by the National Institutes of Health grants EY022639 to CI, EY022158 to PYL, EY030599 to PR, EY001765 core grant to the Wilmer Eye Institute, the RPB Unrestricted Grant (Wilmer), the Research to Prevent Blindness Dolly Green Special Scholar Award to CI, and the Wilmer Eye Institute Seed Fund to CI.

## Additional information

### Funding

| Funder | Grant reference number | Author |
| --- | --- | --- |
| NIH Office of the Director | EY022639 | Carlo Iomini |
| NIH Office of the Director | EY022158 | Peter Lwigale |
| NIH Office of the Director | EY030599 | Panteleimos Rompolas |
| Research to Prevent Blindness | Dolly Green Special Scholar Award | Carlo Iomini |

The funders had no role in study design, data collection and interpretation, or the decision to submit the work for publication.

### Author contributions

Céline Portal, Conceptualization, Data curation, Formal analysis, Validation, Investigation, Visualization, Methodology; Panteleimos Rompolas, Data curation, Formal analysis, Funding acquisition, Investigation, Methodology; Peter Lwigale, Data curation, Formal analysis, Funding acquisition, Investigation; Carlo Iomini, Conceptualization, Data curation, Formal analysis, Supervision, Funding acquisition, Validation, Investigation, Visualization, Methodology, Project administration

### Author ORCIDs

Peter Lwigale (iD) http://orcid.org/0000-0003-1799-4905
Carlo Iomini (iD) https://orcid.org/0000-0001-6483-9540

### Ethics

Animal experimentation: All animal procedures were performed in accordance with the guidelines and approval of the Institutional Animal Care and Use Committee at Icahn School of Medicine at Mount Sinai (protocol number: 18-1561), at Johns Hopkins University (protocol number: MO19M122), and at the University of Pennsylvania (protocol number: 805830).

### Decision letter and Author response

Decision letter https://doi.org/10.7554/eLife.52423.sa1
Author response https://doi.org/10.7554/eLife.52423.sa2

## Additional files

### Supplementary files

• Supplementary file 1. Table: RT-qPCR primers.
• Transparent reporting form

### Data availability

All data generated or analysed during this study are included in the manuscript and supporting files. Individual data points are represented on the figures.

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
