## [Decision Letter]

**Acceptance summary:**

The revised manuscript addresses the criticisms of the reviewers and conclusively supports the main findings – that IHH signaling mediated by cilia present on neural crest cells is essential for cell proliferation and for development of the anterior segment of the eye. The revisions made by the authors to the recent set of editorial suggestions are now taken into account.

**Decision letter after peer review:**

Thank you for submitting your article "Primary cilia deficiency in neural crest cells causes Anterior Segment Dysgenesis" for consideration by *eLife*. Your article has been reviewed by three peer reviewers, and the evaluation has been overseen by Joseph Gleeson as the Reviewing Editor and Marianne Bronner as the Senior Editor. The following individual involved in review of your submission has agreed to reveal their identity: Joon Kim (Reviewer #2).

The reviewers have discussed the reviews with one another and the Reviewing Editor has drafted this decision to help you prepare a revised submission.

Summary:

This is a comprehensive data set exploring the role of cilia and Hedghog (Hh) signalling in the development of the anterior segment of the eye. In this study, Portal et al. provided evidence that primary cilia of neural crest cells are required for the development of the anterior segment of the eye. Using a conditional knockout mouse model, the authors ablated neural crest primary cilia and showed that loss of cilia causes a phenotype similar to that of Anterior Segment Dysgenesis (ASD) in humans. The article focuses on neural crest cells which are ciliated and allow the normal development of the anterior segment of the eye. The authors found that choroidal endothelial cells in the sclera express Indian hedgehog, and the ablation of neural crest primary cilia reduces hedgehog signaling in periocular mesenchyme. This study is novel and adeptly dissects the link between primary cilia and the anterior segment development. The paper is a good example of the value of specific murine models to ask fundamental questions regarding development that other methods are unable to answer at the moment. Defects in this pathway lead to a number of phenotypes including changes in the cornea. The authors show that Ihh is the specific signalling pathway used by periocular mesenchyme cells and that the transcriptions factor Foxc1 and Pitx2 are target genes that are required for normal AS development. The work is well presented with a rich data set and good supplementary data to support findings. This is a significant advance in the field: it provides an explanation for the frequent incidence of ASD in more severe ciliopathies such as Meckel-Gruber syndrome, and offers insights into the pathogenesis of non-syndromic ASD. Reviewers appreciated the challenging nature and high quality of data from intravital microscopy of cilia in corneal stroma and the elegant use of mouse reporter lines. Wnt1-Cre is a very well-established Cre-driver line for neural crest linage tracing, and is the appropriate choice for this study. The three reviewers were mostly positive, but offered some suggestions to increase the impact of the work. But because the reviewers were so positive on the overall study, eLife has included these into the Minor Revisions for your consideration. *eLife* does not consider these to be essential for acceptance of the manuscript for publication, but suggests that you consider to either address these points experimentally or in the text as research questions for the future.

Essential revisions:

1) In Figure 5A, there is a dramatic increase in Ki67 staining intensities in the retina of cKO. Thus the result of Ki67 staining shown in Figure 5 is not convincing. If this is a background signal, the analysis of cell proliferation in the periocular mesenchymal cells may be invalid. BrdU pulse labeling can be an alternative method to analyze cell proliferation.

Minor points:

1) The authors showed that the ablation of primary cilia caused a significant reduction of the expression of Foxc1 and Pitx2 in the cornea (Figure 7A), suggesting the relevance of this study to human ASD. However, additional experiments are needed to support the relevance. Is Foxc1 and Pitx2 downregulation a cell autonomous effect of Hedgehog signal loss? Does Gli directly interact with regulatory regions of Foxc1 and Pitx2 genes? Can Smo agonist SAG rescue the loss of Foxc1 and Pitx2 expression in cKO?

2) Can authors document if conditional Ift88 loss also causes anatomical abnormalities in other post-migratory neural crest derivatives such as cardiovascular structures and mesenchyme in the pharyngeal arches. Presumably these are severe because there is embryonic lethality: do these reiterate ciliopathy phenotypes? Additional information to what is presented in Figure 2A could extend the scope and impact of the study.

3) The important advance in the study is the determination that primary cilia in the periocular mesenchyme mediate canonical Indian Hedgehog (Ihh) signalling. For me, this is fairly conclusive from the data but the most conclusive experiment would be if ex vivo POM would specifically respond to exogenous Ihh ligand rather than Shh. Active recombinant IHH is available commercially. (The may have already considered this approach and found that the protein was inactive in this type of assay). It would also be useful to show Ihh expression at the choroid endothelium by antibody staining.

4) An OMIM ref is given for Biedmond syn type 2 but not for other syndromes mentioned. A ref is also needed for Biedmond syndrome comments on p6

5) Introduction paragraph five. The emphasis of the overall findings here is quite different to the Abstract of the paper.

6) Results section: – why are the current findings of cilia in keratocytes different from previous studies? What could be the reason for the contrast?

7) Figure 1—figure supplement 1 title Ift88 – italics?

8) Figure 6A please give p values for corneal Dhh Ihh and Shh.

9) Figure 4—figure supplement 1 – a schematic would be useful to help orientate the position of the ciliary body of the eye.

10) The title of Figure 6 "During the embryonic development, POM cells of the sclera lose their ability to respond to Ihh produced by choroidal endothelial cells" is somewhat inconsistent with the main text. In the text the authors concluded that "This implies that POM cells of P0 sclera were still able to activate Hh signaling upon ligand stimulation…"

11) Discussion paragraph two first sentence: do you mean "somatic" here?

12) Figure 2B and D: please annotate the ocular structures for non-experts and explain what * shows in B.

13) Figure 3B and C: I did not understand what the graphs were plotting. If these are means of mean values then a t-test would not be appropriate? It would be easier to understand if all individual values were displayed in a dot graph and ANOVA was used to analyze the entire data-set.

14) Figure 3—figure supplement 1: "as many times as necessary to obtain a good segmentation" makes me question the reliability of image segmentation. How was the quality assessed, and would a standard analysis in FIJI be more reproducible?

15) In Figure 5B, the panel shows proliferation data measured by Ki67+ cells, whereas the legend states that BrdU^+^ cells were also counted.

---

## [Author Response]

Essential revisions:1) In Figure 5A, there is a dramatic increase in Ki67 staining intensities in the retina of cKO. Thus the result of Ki67 staining shown in Figure 5 is not convincing. If this is a background signal, the analysis of cell proliferation in the periocular mesenchymal cells may be invalid. BrdU pulse labeling can be an alternative method to analyze cell proliferation.

We thank the reviewers for pointing out this problem. We have realized that for the art work in Figure 5 the intensities of the projections of confocal optical sections were not normalized. In the revised version of Figure 5 we have included in the panels of the enlargements a portion of the retina, the retinal Ganglion Cell Layer (GCL), that does not present Ki67 staining in both control and cKO eyes. Thus, we have utilized the GCL to normalized the intensity of Ki67 positive cells in the rest of the eye tissue. We have corrected this discrepancy and now the intensity of the Ki67 staining in the retina and the periocular mesenchyme is similar in both, control and cKO maximum intensity projections. The quantification didn’t change, since Ki67 positive cells were all counted. In addition, we performed a similar experiment using BrdU pulse labeling (see Figure 5—figure supplement 1). Unfortunately, the BrdU staining requires an acidic treatment that was not compatible with DAPI staining of the nuclei. Thus, BrdU positive cells number was normalized to the selected area of the periocular mesenchyme. The results obtained with the BrdU pulse labeling are consistent with those obtained with Ki67 staining and were added as a supplement to Figure 5.

Minor points:1) The authors showed that the ablation of primary cilia caused a significant reduction of the expression of Foxc1 and Pitx2 in the cornea (Figure 7A), suggesting the relevance of this study to human ASD. However, additional experiments are needed to support the relevance. Is Foxc1 and Pitx2 downregulation a cell autonomous effect of Hedgehog signal loss? Does Gli directly interact with regulatory regions of Foxc1 and Pitx2 genes? Can Smo agonist SAG rescue the loss of Foxc1 and Pitx2 expression in cKO?

Our data suggest a possible signaling axis between the primary cilium and ASD causing genes *Foxc1* and *Pitx2*. We agree with the reviewer that additional studies are needed to demonstrate a possible direct connection between the cilium of periocular mesenchymal cells and the expression of *Foxc1* and *Pitx2.* In this study, we have shown that in the periocular mesenchyme cilia are required to maintain an active Hedgehog signaling cascade. Our findings appear to be consistent with data obtained in other tissues. During sea urchin embryogenesis, disruption of the Hh pathway negatively affects the expression of *Pitx2* gene (Materna et al., 2013, Development, 140 (8):1796-1806; Warner et al., 2016, Dev Biol, 411(2):314-324). Moreover, *Foxc1* expression decreases when the HH signaling is disrupted in the developing heart (Yamagishi et al., 2003, Genes Dev, 17(2):269-281). Thus, the HH pathway could play a role in directly regulating *Foxc1* and *Pitx2* expression. One possibility is that Gli transcription factors directly interact with regulatory elements of *Foxc1* and/or *Pitx2* promoting their expression, as suggested by the reviewer. My laboratory is committed to investigate this possibility and further elucidate the relationship between ciliary elements, including components of the HH signaling, and the *Foxc1* or *Pitx2* genes. In addition, we are considering in vitro experiments that could reestablish normal levels of *Foxc1* or *Pitx2* gene expression in cKO eyes.

2) Can authors document if conditional Ift88 loss also causes anatomical abnormalities in other post-migratory neural crest derivatives such as cardiovascular structures and mesenchyme in the pharyngeal arches. Presumably these are severe because there is embryonic lethality: do these reiterate ciliopathy phenotypes? Additional information to what is presented in Figure 2A could extend the scope and impact of the study.

Several studies have highlighted the essential role of cilia of neural crest cells in craniofacial and heart development. However, in this study we have focused our attention to the role of cilia-mediated signaling in neural crest-derived tissues of the eye.

3) The important advance in the study is the determination that primary cilia in the periocular mesenchyme mediate canonical Indian Hedgehog (Ihh) signalling. For me, this is fairly conclusive from the data but the most conclusive experiment would be if ex vivo POM would specifically respond to exogenous Ihh ligand rather than Shh. Active recombinant IHH is available commercially. (The may have already considered this approach and found that the protein was inactive in this type of assay). It would also be useful to show Ihh expression at the choroid endothelium by antibody staining.

All hedgehog ligands display high sequence conservation and similar biochemical processing, being modified with both palmitoyl and cholesteryl moieties, thus, it is expected that all HH ligands bind to the PATCH receptor(s) (Petrov et al., 2017, Annu Rev Cell Dev Biol, 33:145-168; Kong et al., 2019. Development, 146(10)). Rather, the differences between paralogs appear to be in the timing and location of expression (Ingham et al., 2011, Nat Rev Genet, 12(6):393-406; Jeong and McMahon, 2002, J ClinInvest, 110(5):591-596; Varjosalo and Taipale, 2008, Genes Dev, 22(18):2454-2472). Therefore, we expect that each HH ligand would bind to PATCH receptors of the periocular mesenchyme and activate the HH pathway.

To determine the localization of Ihh in the periocular mesenchyme, we have attempted immunohistochemistry experiments using two different commercial antibodies directed to Ihh: ab52919 (Abcam) and ab39634 (Abcam). Unfortunately, the results were not conclusive. We detected abundant non-specific fluorescent signal in the retina with the antibody ab52919 (see Author response image 1) and the antibody ab39634 gave high background also in the retina (see Author response image 1). Because Ihh is not expressed in the retina (Figure 6A), these staining patterns argued against the specificity of these antibodies in eye tissue. Thus, we do not feel confident to investigate tissue localization of Ihh.

4) An OMIM ref is given for Biedmond syn type 2 but not for other syndromes mentioned. A ref is also needed for Biedmond syndrome comments on p6

We added a reference for the Biemond syndrome 2:

Schachat A.P. and Maumenee I.H., 1982. Bardet-Bield syndrome and related disorders. *Arch Ophthalmol*. 100(2):285-288.

5) Introduction paragraph five. The emphasis of the overall findings here is quite different to the Abstract of the paper.

We have added a sentence to emphasize the results relative to *Foxc1* and *Pitx2*.

6) Results section: why are the current findings of cilia in keratocytes different from previous studies? What could be the reason for the contrast?

Despite corneal endothelial cells and stroma keratocytes share a common neural crest cell precursor, in the adult cornea they are distinct tissues with specific structure and function. Moreover, the corneal endothelial cilium projects into the lumen of the anterior chamber whereas the cilium of keratocytes interacts with collagen fibers of the extracellular matrix. Thus, the cilia in the two cell types may carry out distinct functions. Because at this point we don’t have data supporting this possibility, we prefer not to speculate in this regard.

7) Figure 1—figure supplement 1 title Ift88 – italics?

We apologize for this inaccuracy. *Ift88* has been italicized.

8) Figure 6A please give p values for corneal Dhh Ihh and Shh.

We added the *P* values for corneal Dhh, Ihh, and Shh. For the sake of uniformity in the presentation of the results, non-significant *P* values for Dhh, Ihh, and Shh in the sclera and neural retina have also been added.

9) Figure 4—figure supplement 1 – a schematic would be useful to help orientate the position of the ciliary body of the eye.

We added a schematic of the eye to help orientate the position of the ciliary body and the different ocular structures in the adult eye.

10) The title of Figure 6 "During the embryonic development, POM cells of the sclera lose their ability to respond to Ihh produced by choroidal endothelial cells" is somewhat inconsistent with the main text. In the text the authors concluded that "This implies that POM cells of P0 sclera were still able to activate Hh signaling upon ligand stimulation…"

We thank the reviewers for noticing this discrepancy. We have reformulated the title of the figure and made adjustment in the main test to better express the results that were obtained and presented in the main text. The title of Figure 6 has been changed as follow: "Ihh produced by choroidal endothelial cells is the preeminent HH ligand present in the POM during embryonic development ".

11) Discussion paragraph two first sentence: do you mean "somatic" here?

In the manuscript, we wrote “Mice carrying somatic null mutations…”. Indeed, this was to emphasize that these particular models are constitutive knock-out mice, so that the primary cilium was ablated in all the cells of the mice. In contrast to our model, which is a conditional knock-out since we ablated the primary cilium only in neural crest derived cells.

12) Figure 2B and D: please annotate the ocular structures for non-experts and explain what * shows in B

As suggested, we annotated the different ocular structures in Figure 2B and D (all abbreviations are explained in the figure legend). We apologize for the omission of the meaning of * in B. It has been added in the B section of the legend.

13) Figure 3B and C: I did not understand what the graphs were plotting. If these are means of mean values then a t-test would not be appropriate? It would be easier to understand if all individual values were displayed in a dot graph and ANOVA was used to analyze the entire data-set.

In this instance, ANOVA cannot be applied to analyze our data-set because our observations do not meet the assumptions to perform it. To compare our two variables, the genotype (CTRL / cKO) and the position within the corneal stroma (anterior side / posterior side), we would need a two-way ANOVA. The two-way ANOVA requires having independence of the observations, which means that there is no relationship between the observations in each group or between the groups themselves. In our case, the amount of extracellular spaces in the anterior part and the amount of extracellular spaces in the posterior part are measured on the same corneas (this is also true for the average size of extracellular space). Thus, our observations are not independent, so we cannot use a two-way ANOVA.

In this instance a *t*-tests appears appropriate to analyze our data-set, as stated in the legend of Figure 3. Paired two-tailed Student’s *t*-tests were used to compare the anterior and posterior parts of the stroma in one given genotype (dependent individuals). Unpaired two-tailed Student’s *t*-tests were used to compare the two genotypes (independent individuals).

On Figure 3B and C, individual values were displayed (one symbol/embryo). On Figure 3B, each dot represents the mean value of the amount of extracellular spaces of one embryo, quantified on 5-12 pictures extracted from the anterior and the posterior part of the stroma of the z-stack acquired throughout the cornea. On Figure 3C, each dot represents the mean value of the average size of extracellular spaces of one embryo, quantified on 5-12 pictures extracted from the anterior and the posterior part of the stroma of the z-stack acquired throughout the cornea.

The legend of Figure 3B-C and the corresponding paragraph in Materials and methods have been modified to convey more clarity. Moreover, we added an indication of the cornea polarity (Anterior/Posterior) in Figure 3—figure supplement 1.

14) Figure 3—figure supplement 1: "as many times as necessary to obtain a good segmentation" makes me question the reliability of image segmentation. How was the quality assessed, and would a standard analysis in FIJI be more reproducible?

At first, we tried different segmentations of our z-stacks using Fiji. However, the complexity of the pictures did not allow us to easily segment them. To avoid adding excessive pretreatments of the pictures which could skew the analysis, we opted to adopt “Ilastik”. Differently from Fiji, Ilastik takes into consideration the value of a pixel and the values of its surrounding pixels and it is based on machine learning technology. The legend of Figure 3—figure supplement 1 has been reformulated for more clarity.

15) In Figure 5B, the panel shows proliferation data measured by Ki67+ cells, whereas the legend states that BrdU^+^ cells were also counted.

We apologize for this substitution in the legend. “The number of Ki67^+^/BrdU^+^ cells…” has been replaced by “The number of Ki67^+^/DAPI^+^ cells…” as displayed in Figure 5B.